# Source Apportionment of PM$_{2.5}$ in Montréal, Canada and Health Risk Assessment for Potentially Toxic Elements

Nansi Fakhri[1,2,†], Robin Stevens[2,3], Arnold Downey[2], Konstantina Oikonomou[4], Jean Sciare[4], Charbel Afif*[1,4], Patrick L. Hayes*[2]

[1] EMMA Research Group, Centre d'Analyses et de Recherche, Faculty of Sciences, Université Saint-Joseph, Beirut, Lebanon
[2] Department of Chemistry, Faculty of Arts and Sciences, Université de Montréal, Montréal, Québec, Canada
[3] School of Public Health, Université de Montréal, Montréal, Québec, Canada
[4] Climate and Atmosphere Research Center (CARE-C), The Cyprus Institute, Nicosia, Cyprus

*Correspondence to*: Patrick L. Hayes (patrick.hayes@umontreal.ca); Charbel Afif (charbel.afif@usj.edu.lb)

[†] Current affiliation:

-*EMMA Research Group, Centre d'Analyses et de Recherche, Faculty of Sciences, Université Saint-Joseph, Beirut, Lebanon*
-*Climate and Atmosphere Research Center (CARE-C), The Cyprus Institute, Nicosia, Cyprus*

**Abstract.** Source apportionment of PM$_{2.5}$ was performed using positive matrix factorization (PMF) based on detailed chemical composition data from 24-h filter samples collected over a 3-month period (August-November 2020) at an urban site in Montréal, a Canadian city with a population of approximately 4 million people. This source apportionment study, which examined the main contributing sources to PM$_{2.5}$ using a larger suite of organic molecular markers than other Canadian studies, is the first of its sort in Canada. A focus of this study was on quantifying previously unresolved sources of PM$_{2.5}$ through the inclusion in the PMF analysis of additional organic molecular markers beyond those measured typically by the Canadian government's National Air Pollution Surveillance Program (NAPS). The organic species included in the PMF model were comprised of six n-alkanes, two fatty acids, one dicarboxylic acid, two biogenic secondary organic aerosols (SOA) tracers and hopane. Secondary inorganic aerosols (SIA) and SOA were the dominant components and constituted 39% of the measured PM$_{2.5}$ mass while the local primary anthropogenic sources, namely traffic exhaust, road dust, industrial, and cooking emissions contributed 23%. The chemical transport model GEOS-Chem revealed that ammonium sulfate concentrations in Montréal are strongly influenced by both local sources in Québec and transboundary input from the United States, with the transboundary input exceeding the local emissions for SOA. Co and Cr(VI) presented an elevated cancer risk, highlighting that more attention should be given to these trace metals, which were associated with industrial emissions by the PMF analysis. Furthermore, the results showed that industrial emissions were minor contributors to the total PM$_{2.5}$ mass concentration, but were the largest contributors to Co and Cr(VI) concentrations. Thus, the health hazards associated with this source cannot be entirely established by the PM$_{2.5}$ mass concentration alone. This study highlights that, when evaluating air quality in Montréal and other urban regions, the prioritization of sources for mitigations strategies will diverge

if one considers total PM$_{2.5}$ mass concentration or the concentration of individual particulate-bound contaminants. Furthermore, the large transboundary contribution from the United States to total PM$_{2.5}$ levels suggests that future municipal, provincial and federal monitoring and regulations would be more effective if they focus on specific high-risk contaminants (e.g., Co and Cr(VI) rather than total PM$_{2.5}$.

## 1 Introduction

Outdoor air pollution is a serious threat to human health and is responsible for over 4 million premature deaths worldwide annually (Nansai et al., 2021). Among the various air pollutants, particulate matter (PM) is one of the most problematic and has been classified within Group 1 "carcinogenic to humans" by the International Agency for Research on Cancer (IARC). Particle size is a critical parameter for assessing health impacts, and PM with an aerodynamic diameter less than 2.5 μm (PM$_{2.5}$) represents a greater threat to health than coarser particles as these fine particles can penetrate deeply into the respiratory tract and induce adverse health effects such as respiratory and cardiovascular diseases (Lelieveld et al., 2019). Outdoor air pollution continues to have serious health consequences in Canada (EAR, 2019; Jeong et al., 2011; Bari & Kindzierski, 2016). The government of Canada estimates that approximately 15,300 premature deaths per year are linked to air pollution (Health Canada, 2021). Montréal, a Canadian city in the province of Québec, is the largest city in the province and the second largest in Canada with a population of approximately 4 million people. The city maintains a network of 14 air quality monitoring stations that continuously measure the concentrations of major pollutants (e.g., PM, ozone, sulfur dioxide (SO$_2$), nitrogen oxides (NO$_X$), ozone etc.). These municipal monitoring stations also belongs to Environment and Climate Change Canada's National Air Pollution Surveillance (NAPS) program which is the principal source of ambient air quality data in Canada (EAR, 2020) and provides also speciated PM composition measurements via off-line analyses at a limited number of sites. The resulting data enable a portrait to be drawn of the evolution of pollutant concentrations in Montréal over the years. The concentrations of PM$_{2.5}$ in Montréal's ambient air have been decreasing since 2011. The 3-year average concentration of PM$_{2.5}$ decreased from 9.7 μg/m$^3$ (2011-2013) to 7.3 μg/m$^3$ (2018-2020) (EAR, 2017; 2018; 2019; 2020). However, in September 2021, WHO released revised global air quality guidelines with a recommended PM$_{2.5}$ concentration of 5 μg/m$^3$, meaning that the PM$_{2.5}$ concentrations in Montréal exceed WHO recommendations (WHO, 2021). Thus, it is important to continue to improve air quality in the city.

To effectively reduce PM emissions in the metropolitan area and design effective local PM control strategies, it is necessary to have a better understanding of the dominant emission sources and associated health risk. Over the years several efforts have been made to improve the knowledge about the sources of atmospheric particles in Canada using the receptor-based source apportionment model Positive Matrix Factorization (PMF) and NAPS PM$_{2.5}$ chemical speciation datasets as inputs (Bari & Kindzierski, 2016; Dabek-Zlotorzynska et al., 2011; Jeong et al., 2011; Celo et al., 2021). In Québec, there are six NAPS sites that provide speciated PM$_{2.5}$ measurements, three of which are in the Montréal area namely Châteauneuf (NAPS ID: S50124), Molson (NAPS ID: S50134) and Rivière des Prairies (NAPS ID: S50129). However, samples are collected

once every 3 or 6 days, and operating on this sampling frequency does not allow reliable quantification of the concentration distribution's extremes because this sampling schedule may exclude days of very high or low concentrations from a given source. Another limitation is the lack of data on organic compounds, which can be valuable tracers for certain $PM_{2.5}$ sources (Fadel et al., 2021, Fakhri et al., 2023). To date, investigations of PM concentrations and chemical composition from NAPS sites in Canada have generally focused on determining the elemental (EC) and organic carbon (OC) concentrations as well as the elemental and ionic content of PM.

In this work, $PM_{2.5}$ chemical composition and emission sources as well as potential human health risk associated with trace elements in $PM_{2.5}$ are investigated for an urban site in Montréal over a 3-month period (August-November). In addition to water-soluble ions, elements, EC, and OC, the measurements of $PM_{2.5}$ composition included a large suite of organic molecular markers. This study uses these measurements to explore qualitatively potential sources (e.g., via correlations of elements) before proceeding to a more quantitative approach to source apportionment using PMF. Source apportionment of $PM_{2.5}$ was performed using simultaneously organic and inorganic species in the PMF model. One objective of this work is to investigate previously unresolved PM sources in Montréal, by using some selected organic markers, namely six n-alkanes, hopane, two fatty acids, one dicarboxylic acid, two biogenic secondary organic aerosols tracers and hopane in the PMF model. To our knowledge, this study represents the first time that such extensive composition measurements were included in a source apportionment study in Montréal. Furthermore, the GEOS-Chem chemical transport model was used to evaluate local and long-range contributions to the most significant PM components, providing more information on the origin of PM. Lastly, a health risk assessment model was used to determine the associated risk of the elemental components from the sources identified using the PMF model.

## 2 Method

### 2.1 $PM_{2.5}$ Sampling

Sampling was conducted at an urban site in Montréal from 13 August to 11 November 2020. The sampling site, labeled as MTL, was located on the rooftop of Campus MIL (12 m above ground level) at the University of Montréal (45°31′21″ N, 73°37′14″ W) in the neighbourhood of Outremont. The site is characterized by a high density of residential and commercial premises.

A total of 80 $PM_{2.5}$ filter samples (150 mm quartz-fiber filters, PALL) were collected with a 24 h resolution using a high-volume sampler (CAV-A/MSb, MCV S.A., Spain) operating at 30 $m^3$/h. Filters were baked at 550 °C for 12 h before sampling to eliminate the organic impurities and kept at -20 °C until sampling. Collected filters were also stored at -20 °C until analysis. Organic species and elements were immediately quantified following the field campaign (i.e., within 3 months). Analyses of the water-soluble ions, sugars, OC, and EC were performed a year after the field campaign. Field blank samples were collected by loading filters into the sampler but without operating the sampler's pump. A total number of 8 field blank samples were also analyzed with the same techniques as the sample filters. The concentrations of the species in

the PM$_{2.5}$ samples were corrected by subtracting the average field blank values. Additional QA/QC procedures were applied on the different analysis techniques/protocols used (e.g., determination of detection limits and recovery, as well as validation using certified reference material) (Fakhri et al., 2023). PM$_{2.5}$ mass concentration was determined by weighing the filters before and after sampling using a VWR microbalance with a 1 µg readability (Abdallah et al., 2018, Fadel et al., 2022b).

## 2.2 Chemical analysis

The chemical analyses are detailed in Fakhri et al. (2023) and will only be briefly presented here. Elemental carbon (EC) and organic carbon (OC) were analyzed using the thermo-optical transmission method on a Sunset Laboratory analyzer following the EUSAAR2 (European Supersites for Atmospheric Aerosol Research) protocol (Cavalli et al., 2010).

Major (Al, Fe, K, Mg, Na) and trace (Zn, Ni, Sr, Cu, Ti, Co, Cr, V, Mn, Cd, Mo, Sb, Pb) metals and metalloids were analyzed using an inductively coupled plasma mass spectrometer (ICP-MS) (PerkinElmer, NexION, 300x). In brief, a punch
(18 mm$^2$) was taken from the quartz filters and digested in a mixture of ultrapure HNO$_3$ (4 mL of 67-70% (w/w), Baker ARISTAR ULTRA supplied by VWR International) and HCl (1 mL of 35-38% (w/w), Trace Metal Grade, Fischer Chemical) in a Microwave Reaction System (CEM, Mars-Xpress, MARS 230/60). The oven program was set to an initial temperature ramp reaching 180 °C in 5.5 min with a subsequent holding period of 9.5 min. Matrix-matched ICP-MS calibration standards were prepared using IV-ICPMS-71a (Inorganic Ventures) and quality control standards were prepared
using QCS-27 (High-Purity Standards). Two external calibration methods were used for sample analysis: one for trace metals and metalloids where the calibration range was between 0.05 and 10 µg/L, and the other for the analysis of major metals where the calibration range was between 3 and 500 µg/L. The calibration curves showed good linearity with R$^2$ greater than 0.999. An internal standard solution comprised of Sc, Y, In, and Bi was analyzed along with the samples, with recovery ranging from 90 – 130%. The analytical procedure was validated by considering a certified reference material
NIST-SRM 1648a (Urban particulate matter, National Institute of Standards and Technology, United States of America). Data for a given metal were accepted as quantitative if NIST SRM 1648 recovery was 80 – 110%.

Soluble anions and cations were analyzed by ionic chromatography after extracting a punch (20 mm) of each quartz filter by ultrasonic agitation in 6 mL of MilliQ water (Elga Veolia Purelab Chorus 1, 18.2 MΩ at 25 °C) for 45 min, by means of a Dionex ICS 5000$^+$ instrument. For anion analysis, a Dionex IonPac AS11-HC anion column (2×250 mm), a Dionex IonPac
AG11-HC (2×50 mm) pre-column and a Dionex AERS 500 suppressor were used. Thermo Scientific Dionex KOH with a gradient of 1-45 mM was used as the eluent in the anion analysis with a flowrate of 0.5 mL/min. For cation analysis, a Dionex IonPac CS12 cation column (2×250 mm), a Dionex IonPac CG12 (2×50 mm) pre-column and a Dionex CSRS 300 4 mm suppressor were used during the cation analysis. Thermo Scientific Dionex 12 mM methanesulfonic acid with a flowrate of 0.3 mL/min was used as eluent in the cation analysis.

Sugar alcohols were analyzed by ion chromatography with pulsed amperometric detection (IC-PAD) by means of a Dionex ICS 5000$^+$ instrument using a Dionex CarboPac MA1 column (4×250 mm) and a Dionex CarboPac MA1 (4×50 mm) pre-column. Thermo Scientific Dionex 0.35 M NaOH with a flowrate of 0.6 mL/min was used as eluent in the sugar analysis.

The method used for the organic compounds analysis was described elsewhere (Fadel et al., 2021; Fakhri et al., 2023; El Haddad et al., 2011). Briefly, a punch of the sample filter (73 cm$^2$) was spiked with two internal standards (D50-tetracosane and D6-cholesterol obtained from Sigma Aldrich) followed by an extraction using an accelerated pressurized solvent extraction device at 100 °C and 100 bar (ASE, Dionex 350) and an acetone/dichloromethane mixture (1/1, v/v). After extraction, samples were concentrated to a volume of 200 µL using a constant gentle flow of nitrogen gas. 50 µL of the extracts was then derivatized with the addition of 50 µL of N,O-bis(trimethylsilyl)- trifluoroacetamide (BSTFA) (99%, Sigma Aldrich) with 10 µL of pyridine (catalyst) (99%, Sigma Aldrich) at 70 °C for 2 h. Aliquots of 2 µL of the derivatized extracts were immediately analyzed using a GC-MS in split mode. In addition, aliquots of 2 µL of the non-derivatized fraction were also analyzed using the GC-MS in split mode. The gas chromatograph (Agilent 6890N) was equipped with a HP-5MS UI fused silica capillary column (5%-phenyl, 95%-methylpolysiloxane, 0.25 µm film thickness, and 30 m × 0.25 mm) and interfaced to an ion trap MS with an external electron ionization (EI) source (200 °C, 70 eV). Full scan mode was used in the mass range of 50-550 m/z.

## 2.3 Chemical mass closure

The term "chemical mass closure" refers to the reconstruction of the measured weighed mass using just the chemical composition. It is done by comparing the combined masses of the chemical species to the gravimetric particulate matter mass ($m_{grav}$), wherein the reconstructed PM$_{2.5}$ mass ($m_{chem}$) is defined as the sum of organic matter (OM), EC, crustal matter, sea salt, secondary inorganic aerosol (SIA), and other elements that are not taken into account as minerals (Chow et al., 2015).

A chemical mass closure study was performed using the chemical composition measurements to estimate the contributions of the different components to the total PM$_{2.5}$ mass concentration following the method reported by Fakhri et al. (2023). Briefly, the contribution of sea salt is calculated by summing the six major ions (Sciare et al., 2005):

$$[\text{Sea salt}] = [\text{Na}^+] + [\text{Cl}^-] + [\text{ss} - \text{Mg}^{2+}] + [\text{ss} - \text{K}^+] + [\text{ss} - \text{Ca}^{2+}] + [\text{ss} - \text{SO}_4^{2-}] \quad \text{(Eq. 1)}$$

Ionic constituents such as K$^+$, Ca$^{2+}$, Mg$^{2+}$ and SO$_4^{2-}$ are derived from both marine and non-marine sources. Therefore, it is necessary to discriminate sea salt (ss) from non-sea salt (nss) contributions. Assuming that all sodium ions are of marine origin, the sea salt contribution can be calculated based on sea water composition as shown in Eqs. 2 - 5 (Genga et al., 2017; Sciare et al., 2005). Furthermore, non-sea salt potassium, calcium, magnesium and sulfate (nss-K$^+$, nss-Ca$^{2+}$, nss-Mg$^{2+}$ and nss-SO$_4^{2-}$) are calculated by subtracting the sea-salt fraction (ss-K$^+$, ss-Ca$^{2+}$, ss-Mg$^{2+}$ and ss-SO$_4^{2-}$, respectively) from the total concentration of the ions (K$^+$, Ca2$^+$, Mg$^{2+}$ and SO$_4^{2-}$, respectively).

$$[ss - SO_4^{2-}] = 0.252 \times [Na^+] \qquad (Eq.\ 2)$$

$$[ss - Ca^{2+}] = 0.038 \times [Na^+] \qquad (Eq.\ 3)$$

$$[ss - K^+] = 0.036 \times [Na^+] \qquad (Eq.\ 4)$$

$$[ss - Mg^{2+}] = 0.119 \times [Na^+] \qquad (Eq.\ 5)$$

In the chemical mass closure calculation, we assume that $Na^+$ originates from sea salt, and the ratios between $Na^+$ and other ions ($SO_4^{2-}$, $Ca^{2+}$, $K^+$, and $Mg^{2+}$) in sea salt aerosol are the same as those for seawater. However, it is possible that some $Na^+$ originates from road salt, which is principally composed of NaCl, with a small amount of $CaCl_2$ (Charbonneau, 2006). In this case, the contributions of $SO_4^{2-}$, $K^+$, and $Mg^{2+}$ would be overestimated, and the true concentration of the "road/sea salt" component would be less than that calculated. In the extreme case of the component being derived entirely from road salt, the overestimation in the concentration of this component would be approximately 20%, given the preceding equations. This error is relatively small because $SO_4^{2-}$, $K^+$, and $Mg^{2+}$ have relatively small concentrations in sea salt.

In addition, secondary inorganic aerosol (SIA) is represented by the sum of nss-$SO_4^{2-}$, $NH_4^+$ and $NO_3^-$. To take bound water into account a hydration multiplication factor of 1.29 was applied to convert the dry inorganic concentrations (SIA and sea salt) into hydrated species (Sciare et al., 2005; Genga et al., 2017).

The contribution of crustal matter (CM) (Eq. 6) was estimated by summing the concentrations of aluminum, silicon, calcium, iron, and titanium in their oxide forms (Huang et al., 2014). The coefficients in front of the elements correspond to the additional mass due to oxygen in the minerals. Silicon was not measured in this study and was indirectly determined by multiplying the measured aluminum concentration by a factor of 3.41 (Esmaeilirad et al., 2020). This factor is obtained from the ratio of Si and Al in the Earth's crust following Mason and Moore (1982).

$$[CM] = 2.2\,[Al] + 2.49\,[Si] + 1.63\,[Ca] + 2.42\,[Fe] + 1.94\,[Ti] \qquad (Eq.\ 6)$$

To account for unmeasured O, N, S, and H atoms in OM, the conversion factor (CF) from OC to OM was derived using the equation OM = CF×OC. The method used to calculate the CF sums all the PM components while systematically varying the OM/OC conversion (Genga et al., 2017). To find the optimal CF to calculate OM from OC, the factor was varied from 1.2 to 2.1. The Pearson correlation (R) calculated between the reconstructed $PM_{2.5}$ and the measured mass did not change significantly (0.978-0.979), but the highest correlation and the slope closest to 1 was obtained with CF=1.6. The results of chemical mass closure study are shown in Fig. S5.

## 2.4 Health risk assessment

The United States Environmental Protection Agency (USEPA) recommends using a health risk assessment model to determine the health risks from airborne elements. For each of the three exposure pathways, namely inhalation, ingestion, and dermal contact, the carcinogenic and non-carcinogenic health risks were evaluated for children and adults using the measurements taken at the MTL site. This study analyzes the non-carcinogenic risks of Al, Fe, Cu, Zn, Sb, Co, Cr(VI), Ni, V, Cd, Pb and Mn as well as the carcinogenic risk of Cr(VI), Co, Ni, V, Cd and Pb. The average daily dose (ADD in mg/kg per day) for children (0-17 years) and adults (18-70 years) for the three exposure pathways and the exposure concentration through inhalation ($EC_{inhalation}$ in mg/m$^3$) were calculated following Eqs 7-10 (Fadel et al., 2022a, 2022b; Roy et al., 2019). Each of the parameters used in the various formulas are listed in **Table 1** and were retrieved from USEPA reports (USEPA 2004, 2011). Moreover, since chromium toxicity is attributed to its hexavalent state, Cr(VI) concentration was determined as one-seventh of total Cr (Dahmardeh Behrooz et al., 2021; Hao et al., 2020, Fadel et al., 2022b).

$$ADD_{ingestion} = \frac{C \times IngR \times EXP \times ED \times CF}{BW \times AT} \qquad (Eq. 7)$$

$$ADD_{dermal} = \frac{C \times SA \times AF \times ABS \times EXP \times ED \times CF}{BW \times AT} \qquad (Eq. 8)$$

$$ADD_{inhalation} = \frac{C \times InhR \times EXP \times ED \times CF}{BW \times AT} \qquad (Eq. 9)$$

$$EC_{inhalation} = \frac{C \times EXP \times ED \times ET \times CF}{AT \times 24} \qquad (Eq. 10)$$

**Table 1:** Exposure parameters for children and adults.

| Exposure parameters | | Unit | Adults | Children |
|---|---|---|---|---|
| Concentration of metal in PM$_{2.5}$ | C | ng/m$^3$ (inhalation) and mg/kg (ingestion and dermal) | | |
| Ingestion rate | IngR | mg/day | 50 | 100 |
| Inhalation rate | InhR | m$^3$ per day | 15.6 | 12.3 |
| Exposure frequency | EXP | days/year | 350 | 350 |
| Exposure duration | ED | years | 52 | 17 |
| Conversion factor | CF | kg/mg | 10$^{-6}$ | 10$^{-6}$ |
| Body weight | BW | kg | 70 | 37 |
| Exposed skin area | SA | cm$^2$ | 5700 | 2800 |
| Adherence factor | AF | mg/cm$^2$ | 0.07 | 0.2 |
| Exposure time | ET | h/day | 8 | 8 |
| Dermal adsorption factor | ABS | - | 0.001 for Cd 0.01 for other metals | |
| Averaging time | AT | days | Non-carcinogens: AT=ED × 365 (days) Carcinogens: AT=70 years × 365 | |

The hazard quotient (HQ), which is determined by dividing the ADD from each exposure pathway by a specified reference dose (R$_f$D) (mg/kg per day) for the same exposure route, can be used to measure the non-carcinogenic health risk effects from metals as presented in Eq. 11 (Fadel et al., 2022a, 2022b; Hao et al., 2020). The non-cancer risk refers to the likelihood of developing health issues other than cancer as a result of exposure to chemical pollutants such as asthma, nervous system disorders, cardiovascular and respiratory disorders (Fadel et al., 2022a).

To assess the overall potential for non-carcinogenic consequences produced by multi-element exposure for one exposure pathway, the hazard index (HI$_i$) was calculated as the sum of the HQ$_{ij}$ (i is the exposure pathway which is either inhalation, ingestion, or dermal contact, and j is the targeted compound) (Eq. 12). HI$_{total}$ represents the total hazard index (HI$_{total}$ = HI$_{inhalation}$ + HI$_{ingestion}$ + HI$_{dermal}$). An HI value higher than one implies that the non-cancer risk merits attention and indicates that adverse health effects are likely to occur (Dahmardeh Behrooz et al., 2021).

$$HQ_i = \frac{ADD_i}{R_fD_i} \qquad (Eq.\,11)$$

$$HI_i = \sum HQ_{ij} \qquad (Eq.\,12)$$

The carcinogenic risks (CR) are estimated using Eq. 13-15, which refers to the probability that a person might develop a cancer over a lifetime because of exposure to a carcinogenic chemical (Dahmardeh Behrooz et al., 2021; Fadel et al., 2022a, 2022b). According to the USEPA, a cancer risk value between $10^{-6}$ (one additional case per one million people) and $10^{-4}$ (one in a ten thousand) indicates that the carcinogenic risk is considered as tolerable while a value higher than $10^{-4}$ indicates that a serious risk of cancer exists (Bari and Kindzierski, 2016; Dahmardeh Behrooz et al., 2021). $CR_{total}$ represents the total carcinogenic risk ($CR_{total} = CR_{ingestion} + CR_{inhalation} + CR_{dermal}$).

$$CR_{ingestion} = ADD_{ingestion} \times CSF \qquad (Eq.\,13)$$

$$CR_{inhalation} = EC_{inhalation} \times IUR \qquad (Eq.\,14)$$

$$CR_{dermal} = ADD_{dermal} \times CSF \qquad (Eq.\,15)$$

In the equations above CSF is the cancer slope factor for a chemical in a specific exposure pathway (mg/kg per day) and IUR is the inhalation unit risk (m$^3$/mg) (**Table S1**).

**2.5 Source apportionment**

The USEPA PMF v5.0 software was used to identify and quantify the major emission sources in Montréal. PMF is a multivariate factor analysis tool that decomposes a data matrix X (n x m) into two matrices: source contributions G (n x p) and sources profiles F (p x m), where n is the number of samples, m is the number of species and p is the number of factors or sources. The PMF model requires the concentration data set of the samples and associated uncertainty as inputs. The goal is to solve the chemical mass balance (Eq. 16) between the measured species concentrations and source profiles:

$$x_{ij} = \sum_{k=1}^{p} g_{ik}\, f_{kj} + e_{ij} \qquad (Eq.\,16)$$

where $x_{ij}$ is the concentration of the species j in the i$^{th}$ sample, $g_{ik}$ is the contribution of the k$^{th}$ source in the i$^{th}$ sample, $f_{kj}$ is the relative concentration of species j from the source k, and $e_{ij}$ is the residual of species j in the i$^{th}$ sample. The values of $g_{ik}$ and $f_{kj}$ are adjusted until a minimum value of Q (Eq. 17) for a given number of factors p is found:

$$Q = \sum_{i=1}^{n} \sum_{j=1}^{m} \left(\frac{e_{ij}}{u_{ij}}\right)^2 \qquad (Eq.\,17)$$

where e is the residual value and u is the uncertainty in a measurement. The residual value is the difference between the measured value and the PMF-modeled concentration of each compound. In the present work, samples below the detection limit (DL) were replaced by half of the DL and were given an uncertainty of 5/6 times the detection limit (Polissar et al., 1998). Missing samples were replaced by the median value of that species and were given an uncertainty of 4 times the median value (Polissar et al., 1998). When the concentration was greater than the DL, the uncertainty was calculated according to the USEPA guidelines (USEPA, 2014; Lee et al., 2022; Park et al., 2019):

$$\sqrt{(Concentration \; x \; 0.1)^2 + (0.5 \; x \; DL)^2}.$$

After screening the integrity of the input data, 27 species were included in the PMF model. The overall number of samples (80 samples) and the number of species complies with the ratio of at least 3:1, as proposed by Belis et al., (2019). The species included were OC and EC, major water-soluble ions ($Na^+$, $Cl^-$, $NH_4^+$, $NO_3^-$ and $SO_4^{2-}$) and a selection of elements (Al, Fe, Ti, Cu, Sb, Cd and Co). Levoglucosan was included as a tracer for biomass burning, 17α[H]-21β[H]-Hopane as a tracer for vehicular emissions, fatty acids (hexadecanoic acid and octadecanoic acid) for cooking activities, a set of n-alkanes for biogenic (C27, C29) and anthropogenic emissions (C20, C21, C24, C25), and finally a dicarboxylic acid (oxalic acid) and α-pinene oxidation products (pinic acid and cis-pinonic acid) as tracers for SOA.

All the included species were defined from weak to strong in the PMF model based on their signal-to-noise ratio (S/N). When the S/N ratio was less than 0.2, the PM species were classified as "bad," "weak" when the S/N ratio was between 0.2 and 2, and "strong" when the S/N ratio was greater than 2 (Esmaeilirad et al., 2020). The bad species are excluded from the analysis while the uncertainty for the weak species is tripled. $PM_{2.5}$ was designated as a "total variable" and was automatically classified as "weak". All the included species were successfully modeled by PMF with their concentrations reconstructed accurately and were qualified as "strong" except for nitrate which presented a S/N ratio of 0.9 and was defined as "weak".

The final solution was selected based on several criteria such as (1) comparison of the resulting source profiles against the literature, (2) lack of correlation between the resolved factors, (3) correlation between the predicted vs measured $PM_{2.5}$ concentrations (4) correlation between the modeled and measured species concentrations ($R^2$ higher than 0.8), (5) maximum individual mean (IM) and maximum individual standard deviation (IS) (**Fig. S1**). The $R^2$ between the reconstructed and measured $PM_{2.5}$ mass was 0.87 (slope=0.90) (**Fig. S2**). The robustness of the PMF solution was tested by the two-error estimation method (bootstrap and displacement) as instructed in the PMF manual to ensure the solution was stable (**Table S2**) (USEPA, 2014).

## 2.6 GEOS-Chem Simulations

Simulations were performed using the GEOS-Chem chemical transport model (version 14.0.1, doi:10.5281/zenodo.7271960)
(Bey et al., 2001, Park et al., 2004). GEOS-Chem is driven by assimilated meteorology from the Modern-Era Retrospective
analysis for Research and Applications, Version 2 (MERRA-2), at the NASA Global Modeling and Assimilation Office
(GMAO). The atmosphere was resolved using 47 vertical layers from the surface to 0.01 hPa. The vertical resolution of the
model is about 100 m near the surface, but it becomes coarser at higher altitudes. Boundary conditions were generated using
a global simulation at 2° latitude x 2.5° longitude resolution. A nested grid is then used with 0.5° latitude x 0.625° longitude
resolution, spanning 35° N to 65° N, 90° W to 50° W (**Fig. S3**) in order to include the full province of Québec as well as the
strong source regions of the Great Lakes region and the northeastern US.  We use the recently developed treatment of wet
scavenging described by Luo et al. (2020, 2019), which has been previously shown to yield better agreement for nitrate and
ammonium concentrations over eastern North America.

Biomass-burning emissions were simulated from the Copernicus Atmosphere Monitoring Service Global Fire Assimilation
System (GFAS, Kaiser et al., 2012). We used global emissions from the Community Emissions Data System version 2
(CEDS v2, O'Rourke et al., 2021, Hoesly et al., 2018), except where overwritten by regional emissions inventories, as
described in Keller et al. (2014). These regional emissions inventories included anthropogenic emissions from the US and
most of Canada up to 54° N, provided by the National Emissions Inventory for 2016 (EPA, 2021a), with annual scaling
factors to account for changes in emissions since 2016 (EPA, 2021b). Shipping emissions were provided by CEDS v2, and
aircraft emissions were provided by the Aviation Emissions Inventory Code (AEIC, Stettler et al., 2011, Simone et al.,
2013). Anthropogenic emissions of fine dust aerosols were from the Anthropogenic Fugitive, Combustion, and Industrial
Dust (AFCID) inventory (Philip et al., 2017), while natural dust emissions were calculated according to the Mineral Dust
Entrainment and Deposition (DEAD) parameterization (Zender, 2003). Formation of SOA was parameterized using the
"simple" SOA scheme that treats all organic aerosol as non-volatile, as described in Pai et al. (2020). It has been shown to
reproduce observed organic aerosol concentrations with similar skill to a more complex scheme. GEOS-Chem resolves
mineral dust in four size bins spanning radii of 0.1-1.0, 1.0-1.8, 1.8-3.0, and 3.0-6.0 μm. In this study, the concentration of
dust in particles smaller than 2.5 μm in diameter is estimated by adding 38 % of the concentration of dust in the 1.0-1.8 μm
size bin to the concentration of dust in the 0.1-1.0 μm size bin (Fairlie et al., 2010, Zhang et al., 2013).

GEOS-Chem has previously been evaluated against $NO_2$ concentrations from the USEPA Air Quality System sites (Silvern
et al., 2019), satellite observations of aerosol optical depth, speciated aerosol concentrations from aircraft measurements over
the United States (US) and speciated surface observations from the USEPA Chemical Speciation Network (CSN),
Interagency Monitoring of Protected Visual Environments (IMPROVE), and the Southeastern Aerosol Research and
Characterization (SEARCH) Network (Kim et al., 2015), and ozone concentrations from the Clean Air Status and Trends
Network (CASTNet, Reidmiller et al. 2009). In all cases, GEOS-Chem has been shown to adequately resolve emissions,
atmospheric processes, and large-scale transport. The model performance was evaluated against the observations presented

in this study and good correlation with the observed values was obtained (R=0.24-0.76) (**Table S3**). Some of the indicators are not within the acceptable limits (Abdallah et al., 2018) and this is likely because of the vertical and horizontal resolution of the model. Thus, the PMF results and the model outputs will be compared qualitatively. Furthermore, GEOS-Chem results have been previously used for source contribution analysis similar to the analysis presented in this study (Meng et al. 2019).

In order to examine the sensitivity of air pollutant concentrations to contributions from source regions, a base-case simulation is performed along with three sensitivity simulations, each with anthropogenic emissions from a geographic region turned off: noQC, noCA, and noUS. In the noQC simulation, anthropogenic emissions within the borders of the province of Québec are not allowed. In the noCA simulation, anthropogenic emissions within the borders of Canada are not allowed, except for emissions from the province of Québec (Rest of Canada, RoC). In the noUS simulation, anthropogenic
emissions within the borders of the contiguous US are not allowed. Emissions from shipping and aircraft are removed within the boundaries of the specified region, but emissions from biomass burning or natural sources are not changed. We note that for the purposes of masking emissions, each model grid cell is considered to be entirely within one province or country; the resolution of the provincial or national masks is the same as the model resolution. By calculating the differences in the concentrations of PM components in the sensitivity simulations compared to the base-case simulation, we qualitatively
evaluate the proportions of air pollutants in Québec due to sources within Québec, sources in the RoC, and sources in the contiguous US.

## 3 Results and discussion

### 3.1 PM$_{2.5}$ concentrations

The average PM$_{2.5}$ concentration (and standard deviation) at the MTL site was 4±3 µg/m$^3$. The concentration of PM$_{2.5}$ in all
samples collected at MTL site was lower than the daily standard set by the World Health Organization (WHO) (15 µg/m$^3$) (WHO, 2021) and the Canadian daily standard of 27 µg/m$^3$ (EAR, 2019) (**Fig. 1**). The PM$_{2.5}$ levels at the MTL site can be compared against nearby government monitoring stations to understand if there are large differences in concentrations and how concentrations have changed over recent years in this area of Montréal. The average concentrations at MTL were lower than that reported during the same sampling period at Décarie station (6.6 µg/m$^3$) located ~5 km South-West of the MTL
site near an intersection of two major highways (**Fig. 2**). Given that the Décarie station is strongly impacted by vehicle emissions, this difference is not surprising. In contrast, the average PM$_{2.5}$ concentration at the MTL site is similar to that recorded at the Molson station (NAPS ID: S50134; ~4 km North-East of MTL) (4.7 µg/m$^3$), which is located in a mixed residential/industrial zone for the same sampling period. Furthermore, there is a general decreasing trend between 2017 and 2020 in PM$_{2.5}$ levels at the government monitoring stations. Since the concentration of PM$_{2.5}$ in 2020 was not too different in
comparison with the previous years (2018 and 2019), the sources of PM$_{2.5}$ identified in this study are likely to be similar to other years. It is important to mention that during our sampling period, Montréal was in partial lockdown where public spaces (e.g., bars, gyms, cinemas, museums, libraries and casinos) were closed due to the possibility of a second wave of the

COVID-19 pandemic. Primary and some secondary schools were opened during that period. While these considerations suggest that the results presented here are also applicable to pre-and post-pandemic conditions, further studies are needed

before generalizing the results of this study to other periods.

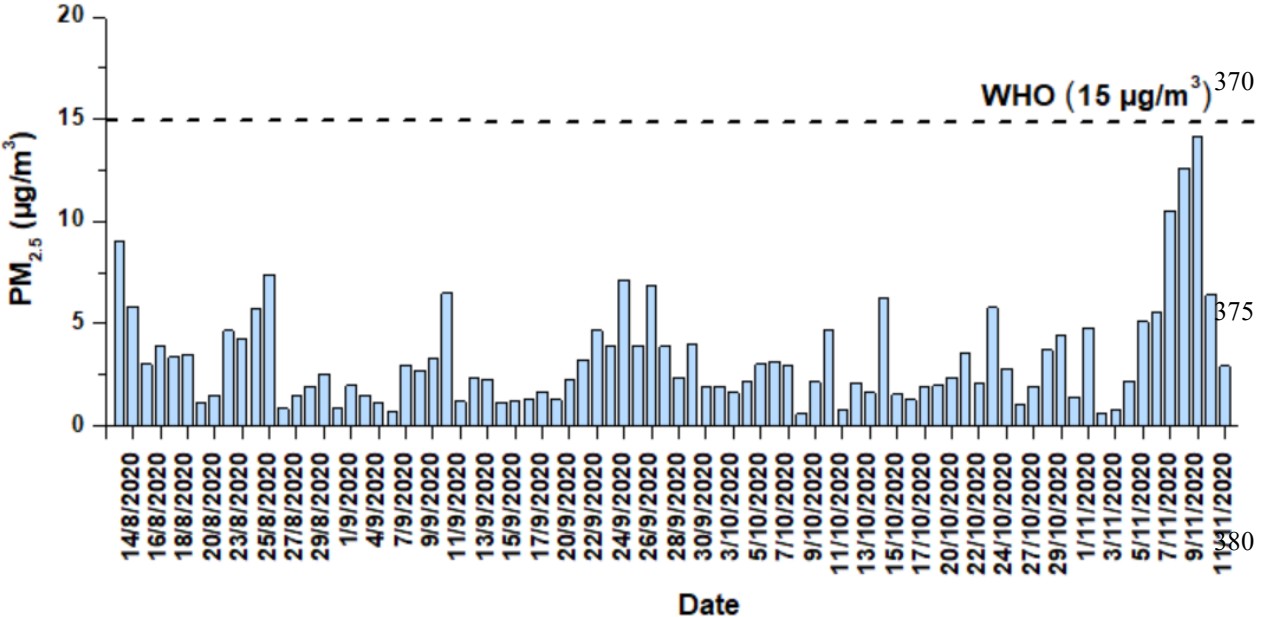

**Figure 1.** The temporal variation of PM$_{2.5}$ concentrations for the sampling period (13 August to 11 November 2020).


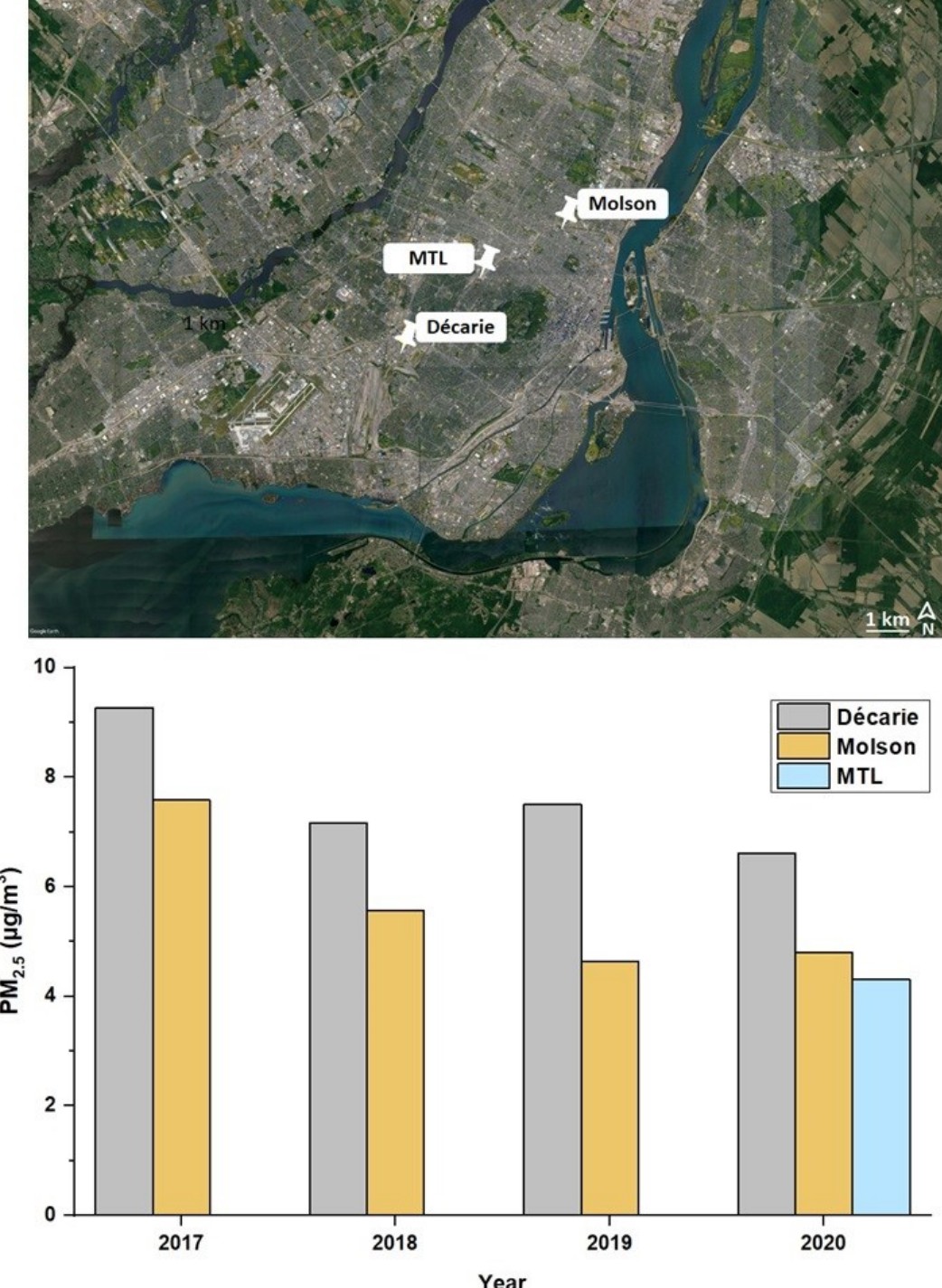

**Figure 2.** PM$_{2.5}$ concentrations for the sampling period (13 August to 11 November 2020) at MTL, and for Décarie and Molson sites between 2017 and 2020 along with the location of the sampling sites (Google Earth®). For Décarie and Molson sites, the data correspond to the same dates of the year (13 August to 11 November).

## 3.2 Carbonaceous matter

The average OC and EC concentrations (± standard deviation) were $1.31\pm0.76$ μg/m$^3$ and $0.17\pm0.15$ μg/m$^3$, respectively (**Table 2**). While EC originates from combustion sources and has a primary origin, OC can be directly emitted from combustion of fossil fuels, biomass burning, cooking activities, and can also be formed in the atmosphere through chemical reactions (Hallquist et al., 2009). The OC/EC ratio is a useful diagnostic ratio that provides information on the sources in PM$_{2.5}$. An OC/EC ratio ranging between 0.3 and 1 was reported for diesel vehicles, between 1.4 and 5 for gasoline operated vehicles while a larger OC/EC ranging between 4.1 and 14.5 for biomass burning (Khan et al., 2021; Salameh et al., 2015). In this study, the ratio of OC/EC was $7.4\pm3.1$, which is larger than those associated with vehicular emissions and consistent with the presence of biomass burning. Indeed, wood burning may be an important source of air pollutants in Québec and wood-burning stoves and fireplaces are used in both residential and businesses settings (e.g., pizzerias and bagel factories). The production of secondary organic carbon will also increase OC/EC ratios beyond values measured for primary sources. Indirect methods have been developed for identifying primary (POC) or secondary organic carbon (SOC) (Shivani et al., 2019; Calvo et al., 2008; Joseph et al., 2012).

$$SOC = OC_{total} - EC \times \left(\frac{OC}{EC}\right)_{min} \quad (Eq.\,18)$$

One such method is summarized in Eq. 18, OC$_{total}$ is the measured OC and (OC/EC)$_{min}$ is the minimum ratio observed in the samples. Additional information on the calculation method is included in the supplementary material. Using this method, the average contribution of SOC to the total OC and the standard deviation was $61\pm15\%$ at the MTL site indicating a strong SOC contribution.

## 3.3 Organic species

Some organic compounds are specific to a certain source and have been used in source apportionment studies as molecular markers. The average concentrations of selected organic compounds in PM$_{2.5}$ measured at the MTL site are summarized in **Table 3**. Levoglucosan, generated by the pyrolysis of cellulose and often utilized as a particular marker of biomass burning (Simoneit, 2002), was the most abundant compound among the organic tracers examined. Due to its presence in lubrication oil used in both gasoline and diesel vehicles, 17α(H)-21β(H)-Hopane is a specific marker of traffic emissions, and it was the least prevalent measured organic species in the PM$_{2.5}$ (El Haddad et al., 2009; Fadel et al., 2021). Arabitol and mannitol were well correlated (R=0.98; *p<0.001*) and are commonly described as markers for primary biogenic emissions, more specifically with fungal spores (Petit et al., 2019). Alkanes, specifically acyclic saturated hydrocarbons, can originate from both biogenic and anthropogenic sources. The n-alkane with the highest concentration was C29. In the literature, low molecular weight n-alkanes (<C27) are primarily associated with traffic emissions, whereas high molecular weight n-alkanes (C27, C29, C31) are associated with plant detritus because they are abundant in the epicuticular wax of plants (Rogge et al., 1993a, 1993b).

To further evaluate the contribution of the anthropogenic and the biogenic sources to n-alkanes, the carbon preference index (CPI) was calculated (Fadel et al., 2021; Esmaeilirad et al., 2020). Two CPI parameters were adopted: the Overall CPI for the whole range of n-alkanes (C15-C30), and the High CPI for the higher molecular weight n-alkanes (C25-C30). The

detailed description of the calculation method is included in the supplementary material. The average Overall CPI value was 0.86±0.21, indicating the contribution of petrogenic sources. The High CPI values during the entire campaign were between 0.74 and 2.81, with a majority   of the measurements in the anthropogenic range and the rest in the mixed anthropogenic/biogenic range (**Fig. 3**). The average High CPI was 1.56±0.48, indicating that larger n-alkanes at the MTL site are predominately anthropogenic with a lesser biogenic contribution.


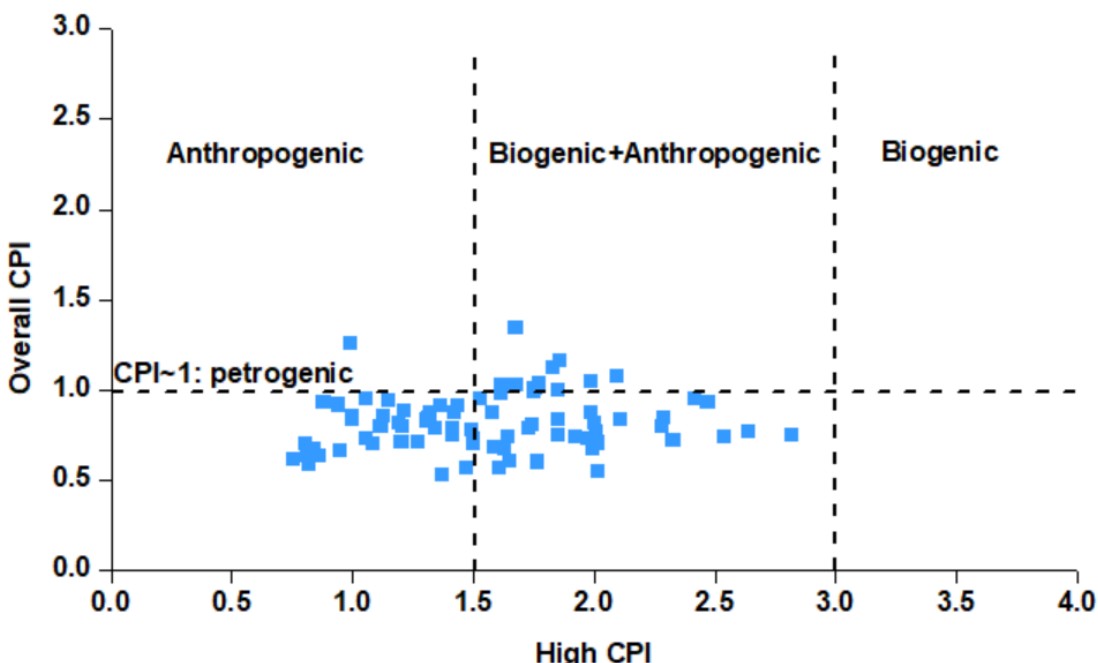

**Figure 3.** The n-alkanes source identification using the carbon preference index (CPI).



**Table 2:** Concentrations of major and trace components of $PM_{2.5}$ measured at MTL site.

| | Mean | Stdev | Range |
|---|---|---|---|
| | | | µg/m³ |
| **OC** | 1.31 | 0.76 | 0.64-4.23 |
| **EC** | 0.17 | 0.15 | 0.05-0.85 |
| **$SO_4^{2-}$** | 0.56 | 0.40 | 0.09-2.34 |
| **$NO_3^-$** | 0.18 | 0.29 | 0.01-2.21 |
| **$NH_4^+$** | 0.36 | 0.23 | 0.03-1.32 |
| **$Na^+$** | 0.04 | 0.02 | 0.003-0.13 |
| **$Cl^-$** | 0.03 | 0.03 | 0.003-0.13 |
| **$Mg^{2+}$** | 0.01 | 0.007 | 0.001-0.06 |
| **$K^+$** | 0.03 | 0.03 | 0.002-0.19 |
| **$Ca^{2+}$** | 0.11 | 0.21 | 0.019-1.90 |
| | | | ng/m³ |
| **Na** | 109.44 | 54.52 | 33.62-335.33 |
| **Mg** | 42.62 | 44.04 | 10.79-380.38 |
| **Ti** | 2.57 | 4.34 | 0.13-36.11 |
| **Al** | 111.17 | 62.45 | 4.93-405.3 |
| **K** | 69.68 | 36.63 | 21.72-283.02 |
| **Mn** | 2.97 | 2.76 | 0.79-24.72 |
| **Fe** | 76.07 | 101.66 | 0.69-932.46 |
| **Zn** | 39.53 | 38.52 | 14.04-240.40 |
| **Ni** | 1.21 | 0.93 | 0.33-6.35 |
| **Sr** | 1.05 | 1.41 | 0.25-11.95 |
| **Cu** | 15.26 | 19.45 | 0.44-101.8 |
| **Co** | 3.81 | 5.76 | 0.026-27.8 |
| **Cr** | 3.05 | 0.12 | 0.05-0.72 |
| **V** | 0.31 | 0.35 | 0.03-2.68 |
| **Cd** | 0.13 | 0.17 | 0.0004-1.21 |
| **Mo** | 0.30 | 0.13 | 0.13-0.99 |
| **Sb** | 0.25 | 0.21 | 0.04-1.20 |
| **Pb** | 5.66 | 5.41 | 0.63-33.4 |

**Table 3:** Average concentrations (ng/m$^3$) of organic compounds in PM$_{2.5}$.

| | Mean | Stdev | Range |
|---|---|---|---|
| 17α(H)-21β(H)-Hopane | 0.18 | 0.01 | 0.06-0.70 |
| **n-Alkanes** | | | |
| Tetradecane (C14) | 0.58 | 0.31 | 0.20-1.73 |
| Pentadecane (C15) | 0.48 | 0.66 | 0.28-4.27 |
| Hexadecane (C16) | 0.75 | 0.37 | 0.34-2.04 |
| Heptadecane (C17) | 0.71 | 0.67 | 0.22-5.20 |
| Octadecane (C18) | 0.99 | 1.69 | 0.18-8.29 |
| Nonadecane (C19) | 0.46 | 0.24 | 0.17-1.26 |
| Eicosane (C20) | 1.82 | 1.20 | 0.73-7.90 |
| Heneicosane (C21) | 0.99 | 0.35 | 0.37-2.14 |
| Docosane (C22) | 0.96 | 1.15 | 0.37-8.49 |
| Tricosane (C23) | 1.02 | 0.42 | 0.40-2.16 |
| Tetracosane (C24) | 2.24 | 1.61 | 0.82-11.16 |
| Pentacosane (C25) | 1.97 | 1.02 | 0.83-4.97 |
| Hexacosane (C26) | 1.51 | 0.84 | 0.48-3.96 |
| Heptacosane (C27) | 2.52 | 2.89 | 0.74-23.82 |
| Octacosane (C28) | 1.79 | 2.84 | 0.45-23.20 |
| Nonacosane (C29) | 3.84 | 3.78 | 0.73-3.37 |
| Triacontane (C30) | 1.12 | 1.41 | 0.12-7.07 |
| Hentriacontane (C31) | 1.56 | 0.96 | 0.34-6.72 |
| **Sugars** | | | |
| Levoglucosan | 33.72 | 6.45 | 6.45-126.40 |
| Mannosan | 1.03 | 0.94 | 0.15-4.59 |
| Mannitol | 2.14 | 3.22 | 0.23-18.91 |
| Arabitol | 3.14 | 4.29 | 0.29-39.00 |
| Glucose | 2.92 | 3.01 | 0.17-24.47 |
| **Fatty acids** | | | |
| Tetradecanoic acid | 4.17 | 1.40 | 1.66-7.97 |
| Hexadecanoic acid | 51.12 | 13.09 | 28.39-87-43 |
| Octadecanoic acid | 37.06 | 9.41 | 18.05-64.90 |
| Oleic acid | 4.43 | 1.65 | 2.51-13.05 |
| **Dicarboxylic acids (DCAs)** | | | |
| Oxalic acid (diC2) | 7.79 | 1.58 | 0.28-15.03 |
| Adipic acid(diC6) | 1.60 | 4.18 | 0.31-3.42 |
| Azelaic acid (diC9) | 5.93 | 2.18 | 1.12-15.01 |
| **Biogenic SOA Tracers** | | | |
| Pinic acid | 4.73 | 1.78 | 0.21-8.93 |
| Cis-pinonic acid | 3.17 | 1.71 | 0.62-9.73 |

## 3.4 Elemental composition

The relationship between trace metals provides qualitative information on the sources of the measured elements. A significant correlation ($R>0.95$, $p<0.05$) was observed among Ti, Fe, Mg, V, K, Mn and Na, indicating a common crustal source. Numerous studies have analyzed the chemical composition of traffic-related PM and have reported that Cu, Sb and Cd are linked to non-exhaust emissions, more precisely from vehicular brake wear (Thorpe and Harrison, 2008; Lin et al., 2015; Pio et al., 2013; Mancilla et al., 2012; Pant et al., 2013). In this study, a correlation of $R=0.82$ ($p<0.05$) was found

between Sb and Cd indicating that Sb and Cd originate from the same sources, most probably brake wear. Studies in other Canadian cities, namely Toronto and Vancouver, have also associated the elements Sb and Cd with non-exhaust emissions from road traffic (Celo et al., 2021). No correlation was found between Cu and the elements Cd and Sb ($R<0.01$, $p<0.05$); indicating that brake wear debris was not an important source of Cu in Montréal.

     Additionally, Co was correlated with Cr ($R=0.79$, $p<0.01$) and Cu ($R=0.86$, $p<0.05$), suggesting a common source from

industrial emissions, similar to observations in Edmonton, Canada (Bari and Kindzierski, 2016). The Canadian Copper Refinery and Suncor Energy refinery can be identified as potential sources of Cr and Co in the Montréal region based on the National Pollutant Release Inventory (NPRI) data published by the Government of Canada (NPRID, 2022). It is important to mention that Cu, Cr and Co could also originate from other sources such as traffic-related emissions and coal combustion (Riffault et al., 2015; Bari & Kindzierski, 2016; Thorpe & Harrison, 2008; Celo et al., 2021). However, these possibilities

seem less important given the correlations between the elements as well as the lack of coal combustion in the province of Québec. Lastly, no correlation was found between Zn, Pb and Sb with $Cl^-$ ($R<0.09$, $p<0.05$) revealing that incinerators are not a potential source of these trace elements (Riffault et al., 2015; Rahn and Huang, 1999).

## 3.5 Water-soluble ions

Among the water-soluble ions, $SO_4^{2-}$ presented the highest concentration followed by $NH_4^+$, $NO_3^-$, $Ca^{2+}$, $Na^+$, $Cl^-$, $K^+$, and the species with the lowest concentration was $Mg^{2+}$. The secondary inorganic aerosol components (sulfate, ammonium, and nitrate) accounted for 83% of the total water-soluble ions and 32% of the total $PM_{2.5}$ mass (**Fig. S5**). The presence of secondary sulfate is supported by the strong correlation between sulfate and ammonium with a Pearson coefficient of 0.90 ($p<0.005$). This result, and the weak correlation between ammonium and nitrate ($R=0.27$, $p<0.002$), suggests that a

significant fraction of ammonium in $PM_{2.5}$ was associated with ammonium sulfate with a limited amount of ammonium nitrate. Nitrate concentrations were higher in the end of October and November in comparison with the warmer months (**Fig. S6**) consistent with a shift in equilibrium partitioning from gas-phase nitric acid to the particle-phase nitrate due to colder temperatures (Geng et al., 2013; Mantas et al., 2014). The Total Ammonium/Total Sulphate (TA/TS) molar ratio was used to evaluate if the site conditions were ammonia-rich ($>2$) or ammonia-poor ($<2$) (Joseph et al., 2012; Remoundaki et al., 2013).

The TA/TS molar ratio was 1.24, highlighting that the concentrations of ammonium are not sufficient to neutralize the

concentrations of sulfate and nitrate. Therefore, the latter anions could be associated to different cations such as $Na^+$, $Mg^{2+}$, $K^+$ and $Ca^{2+}$ and the formation of salts with these cations (e.g., $NaNO_3$, $MgSO_4$). Moreover, when considering the neutralization of all the cations ($Na^+$, $Mg^{2+}$, $NH_4^+$, $K^+$, $Ca^{2+}$) by all the anions ($SO_4^{2-}$, $NO_3^-$, $Cl^-$), the scatter plot (**Fig. S7**) presented a slope of 0.98, indicating a charge balance between the anions and cations.


## 3.6 Source apportionment of PM$_{2.5}$

### 3.6.1 Source identification

The source profiles obtained from the PMF model are presented in **Fig. 4**. Eleven sources were identified at MTL site namely SOA, secondary inorganic aerosol, crustal dust, marine, biomass burning, cooking, traffic exhaust, road dust,
industrial, plant wax and biogenic SOA. The species with the highest loadings were used to identify each factor, and comparison to source profiles found in the literature served as confirmation.

The SOA factor was distinguished with high loading of oxalic acid (75% of the total concentration of oxalic acid) and contributed to 22% of the PM$_{2.5}$ mass concentration (Petit et al., 2019). Oxalic acid is a byproduct of oxidation from various precursors including biogenic (e.g., isoprene) and anthropogenic (e.g., cycloalkanes) compounds, and it is also generated
from the photochemical oxidation of larger acid homologues (Srivastava et al., 2019).

The secondary inorganic aerosol (SIA) factor was distinguished by the presence of the water-soluble ions $SO_4^{2-}$ (58%) and $NH_4^+$ (59%). This factor accounted for 17% of the total PM$_{2.5}$. No trend was observed in the time series of this factor and this source can originate from both local emissions and long-range transport (LRT) originating from the rest of Canada (e.g., Ontario) and the United States (Seinfeld and Pandis, 2016; NPRID, 2022). Aluminum production and industrial processes
related to metallurgy and all other minor sources contributed to ~ 90 000 tons of $SO_2$ in Québec in 2020, which consists of 15% of the total Canadian emissions (583 008 tons) (NPRID, 2022). Although based on the weak correlation between sulfate and Al and strong correlation of Al with crustal elements, we believe that aluminium production is not an important source of particulate aluminum at our site.

The crustal dust factor was identified by high loadings of Al (68%), Fe (76%) and Ti (63%). This factor accounted for 12%
of the total PM$_{2.5}$ and is likely associated with crustal dust sources such as wind-driven resuspension, construction and agricultural activities (Bari and Kindzierski, 2016). All of these crustal dust sources are plausible given that construction is omnipresent in Montréal, and the surrounding St. Lawrence Valley has a large amount of agriculture.

A marine factor was characterized by the ions $Na^+$ (46%), $Cl^-$ (69%) and $NO_3^-$ (30%), contributing to 11% of the PM$_{2.5}$. The $Cl^-/Na^+$ calculated for this factor was 0.95, which is lower than the ratio of 1.80 reported for fresh sea salt and is indicative of
aged sea salt (Petit et al., 2019; Seinfeld and Pandis, 2016). The presence of high nitrate loading in the profile further confirms the presence of aged marine salt. The observed chloride depletion is due to the reaction of nitric and sulfuric acid with NaCl particles (Seinfeld and Pandis, 2016). While the factor has been tentatively identified as "marine", there is some

evidence that this factor may originate, at least partially, from road salt. The marine factor exhibits relatively high concentrations for multiple wind directions including from the west and southwest (Fig. S10), and thus, the marine factor pollution rose resembles to some extent that of road dust. It is also notable that the marine factor exhibits its highest concentrations in November when minimum temperatures were below freezing, and some snowfall occurred. Based on these findings, we suggest that further work is needed to evaluate the contribution of road salt to $PM_{2.5}$ in Montréal. Biomass burning was identified by high loadings of levoglucosan (70%) (Fadel et al., 2023). The OC/EC in this factor was 8.4, consistent with biomass burning (Khan et al., 2021). This source accounted for 9% of the total $PM_{2.5}$ mass. Levoglucosan is a major pyrolysis product of cellulose and hemicellulose (Simoneit, 2002), and has been used as a molecular marker of biomass burning aerosols in several source apportionment studies (Gadi et al., 2019; Shivani et al., 2019). No trend was observed in the time series of this factor with season or temperature, indicating that levoglucosan originates from both residential burning and forest fires.

A cooking emissions factor was identified based on the contribution of hexadecanoic (65%) and octadecanoic acids (68%) in the profile and accounted for 9% of the total $PM_{2.5}$ mass. These carboxylic acids have been used in source apportionment studies to distinguish cooking activities (Gadi et al., 2019; Lv et al., 2021; Shivani et al., 2019).

The traffic exhaust factor was identified by the presence of 17α[H]-21β[H]-Hopane and lower molecular weight n-alkanes (C20 to C25) and accounted for 6% of total $PM_{2.5}$ mass concentration. Hopanes are specific markers of traffic emissions due to their presence in lubrication oil (Rogge et al., 1993a; Schauer et al., 2002). Furthermore, dynamometer tests results showed that the most abundant n-alkanes were C20 and C21 for diesel vehicle emissions and C24 – C25 for gasoline-powered vehicle emissions (Rogge et al., 1993a), and these compounds are commonly associated with motor vehicle emissions. This traffic exhaust factor contributed around 72% of the hopane mass in the model, 59% of the C24 mass and 62% of the C25 mass. We suggest that the traffic emission factor is comprised principally of primary exhaust emissions given the lack of secondary tracers in the factor profile.

The road dust factor was characterized by high loadings of Cd (69%) and Sb (58%) and accounted for 2% of $PM_{2.5}$. These elements are linked to non-exhaust vehicle emissions, particularly from brake-wear debris (Thorpe and Harrison, 2008; Lin et al., 2015). Upon close examination of the PMF factor profiles, one notices some very small mixing of the traffic exhaust, road dust and crustal dust factors, which is a limitation of this study. However, the amount of mixing is very minor and should not impact the conclusions drawn from these results. In this study, PMF allocated 76% of Fe and 68% of Al to the crustal dust factor. In comparison, only 2% of Fe was allocated to the road dust factor while the amount of Al was 4%. Moreover, for the traffic exhaust factor, these values were 2% and 6% for Fe and Al, respectively. It is also possible that these metals are truly associated with the identified sources. Previous literature has found Fe- and Al-containing particles in vehicle exhaust (Golokhvast et al., 2015; Wang et al., 2021). It is also logical that road dust would contain some crustal elements. The industrial emissions factor was dominated by Cu and Co and contributed 70% and 90% of the mass of these elements, respectively, and accounted for 6% of $PM_{2.5}$. These elements are emitted from metal-industry related sources and

coal combustion (Riffault et al., 2015; Bari and Kindzierski, 2016; Sharma and Mandal, 2017), but coal is not used for electricity generation in Québec and thus industrial sources seem more likely to be responsible for this factor.

The plant wax factor was characterized by high loadings of C27 and C29 with 55% of the C27 and 58% of the C29 apportioned to this factor (Fadel et al., 2023). These compounds have been previously linked to primary biogenic emissions
(Rogge et al., 1993b). This factor contributed to 2% of the total $PM_{2.5}$.

Finally, the biogenic SOA factor accounted for 75% of the measured pinic acid and 66% of the measured pinonic acid and accounted for 4% of $PM_{2.5}$. α-Pinene is one of the most atmospherically important compounds in the monoterpene family, and pinic acid and pinonic acid are derived from the photooxidation of α-pinene with ozone ($O_3$) and hydroxyl (OH) radicals (Fadel et al., 2021). Although it is uncommon to see these species included in PMF models, doing so provides insight into
how much biogenic SOA contributes to $PM_{2.5}$ mass concentration. A small percentage of Cu, Sb and Fe are attributed to Biogenic SOA. Specifically, Fe, Sb, and Cu were 5%, 6%, and 5%, respectively. Fe was allocated in much higher proportion to the crustal dust factor, Sb to the road dust factor, and Cu to the industrial factor. Our PMF analysis is consistent with a study reported by Fadel et al. (2023). Fadel and coworkers also included biogenic SOA tracers in the PMF analysis and in their biogenic SOA profile one also notices small amounts of metals/elements.


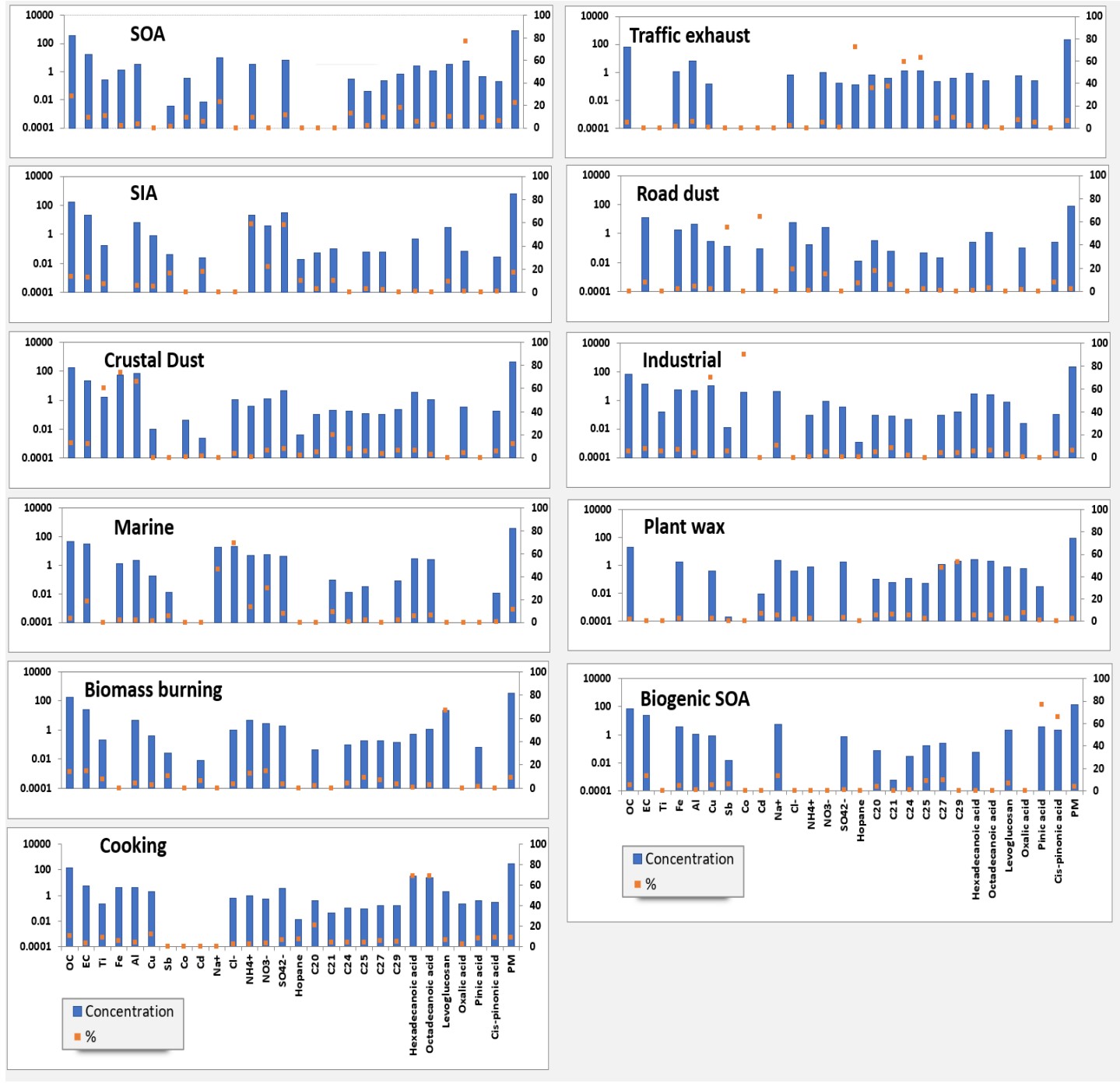

**Figure 4.** Profiles of the eleven factors identified from the PMF model. The left axis corresponds to the concentration of each species (blue bars) and the right axis corresponds to the percentage of each species (orange markers). Units of concentration are ng/m$^3$.


### 3.6.2 Source contribution to PM$_{2.5}$

One of the originalities of the present study relies on the identification of primary and secondary biogenic and anthropogenic sources based on the inclusion of selected organic markers in the PMF model namely n-alkanes, anhydrosugars, a hopane, fatty acids, dicarboxylic acids, and biogenic secondary organic aerosols tracers. Of a total of 11 sources, the addition of organic markers allowed the identification of 4 of them, namely SOA, cooking, plant wax, and biogenic SOA.

When the PM$_{2.5}$ sources identified in this study were compared to other source apportionment studies conducted in major urban areas in Canada (Bari & Kindzierski, 2016; Jeong et al., 2011; Celo et al., 2021) where NAPS PM$_{2.5}$ chemical speciation datasets were used as inputs, this work resolved additional primary and secondary biogenic and anthropogenic sources due to the inclusion of a large suite of organic markers in the PMF model that were not available previously. The additional sources identified, namely SOA, BSOA, plant wax, cooking emissions are also rarely apportioned in the literature (Gadi et al., 2019; Lv et al., 2021).

There are also important differences between the source profiles for vehicular emissions in this study versus previous source apportionment studies conducted in Canada. In Montréal for example, Jeong et al. (2011) identified a traffic emission factor based on the contributions of OC, EC and oxalate. On the other hand, the traffic factor was identified in Edmonton based on the contribution of Ba, Sb, EC, Cu and Co (Bari and Kindzierski, 2016). Since trace elements and carbonaceous matter could be emitted from a variety of sources, this study refined the evaluation of vehicular sources by incorporating two source-specific organic tracers in the PMF model, namely n-alkanes and a hopane. This allowed the differentiation of exhaust and non-exhaust emissions.

The PMF results indicate that SOA and SIA were the largest contributors to fine PM and together constituted 39% (1.68 µg/m$^3$) of the measured PM$_{2.5}$ mass (**Fig. 5**). The primary local urban anthropogenic sources, namely traffic exhaust, road dust, industrial and cooking emissions contributed to 23% (0.99 µg/m$^3$) of the measured PM$_{2.5}$ mass. These sources along with crustal dust, biomass burning, and plant wax comprise the primary aerosol fraction that is 44% (1.94 µg/m$^3$) of the measured PM$_{2.5}$ mass. It should also be noted that residential burning likely contributes to the biomass burning factor, but this factor is not included with the primary local urban anthropogenic sources listed above since it is not possible to distinguish residential burning from wildfires in our PMF analysis. Pollution rose plots (Figure S10) were used to analyze the correlations between wind direction and factor concentrations by plotting in a polar graph the frequency of different concentrations of a factor as a function of wind direction. Such analyses provided information on the potential local origin of the factors. Additional information is included in the supplementary material.

The chemical transport model GEOS-Chem was used to qualitatively evaluate the relative contributions from three different source regions, namely Québec, RoC, and United States to the sources (**Fig. S4**). To link the modelling results to the PMF factors, we focus on modelled concentrations of SOA, of dust in particles smaller than 2.5 µm in diameter, and of the sum of NH$_4$ and SO$_4$. As most of the NH$_4^+$ and SO$_4^{2-}$ was observed to be in the SIA PMF factor, we expect the sum of the simulated

concentrations of $NH_4$ and $SO_4$ from GEOS-Chem have similar sources as the SIA PMF factor. Similarly, the concentrations of SOA from GEOS-Chem would be analogous to the sum of the SOA and biogenic SOA factors, although we note that only anthropogenic sources were altered in the sensitivity simulations. The dust species in GEOS-Chem comprises not only road dust and crustal dust, but all elements not included in the other model aerosol species (primary and secondary organic aerosol, elemental carbon, sulfate, nitrate, ammonium, and sea-spray aerosol). It is therefore most comparable to the sum of the crustal dust, road dust, and industrial PMF factors. Together, the SIA, SOA, biogenic SOA, crustal dust, road dust, and industrial PMF factors comprise 61% of the total observed $PM_{2.5}$ mass.

According to the findings of the chemical transport modelling, Montréal air pollution concentrations are influenced by all three of the regions that were considered. As presented in **Fig. 6**, the US makes an important contribution to SOA and ammonium sulfate concentrations. The concentrations of SOA dropped by 22% between the base case and the sensitivity simulation without Québec emissions and by 36% when the US emissions were not included. The concentrations of ammonium sulfate decreased by 33% when US emissions were excluded and by 35% when emissions from Québec were excluded. Thus, GEOS-Chem simulations reveal an important contribution from US emissions to ammonium sulfate and SOA concentrations. Regarding the sources of sulphate in Québec, it is somewhat surprising that the contributions from the province and the US are essentially the same (35% vs. 33%) given that there are no coal-fired powerplants in Québec while coal is still used at some powerplants in the US. This finding indicates that it is important to consider other sources that contribute to sulphate regionally. Specifically, aluminum production is a major industry in Quebec that emits large amounts of $SO2$ (NPRID, 2022). Nearly, 70% of North American aluminum is produced in Québec. In addition, other industries involving smelting and metallurgy in Québec emit $SO_2$. When also considering the recent decreased use of coal in the US (USEIA, 2022), these alternate sources of sulphate appear to be relatively important in Québec.

On the other hand, anthropogenic dust emissions from Québec presented the highest apportioned contribution to total dust concentrations among the three regions studied, and the concentrations dropped by 16% when emissions from Québec were excluded and by 10% when US emissions were excluded. The sum of the anthropogenic fractions is 30% for dust in GEOS-Chem, which is close to the ratio of the $PM_{2.5}$ mass in the industrial and road dust PMF factors to the sum of the industrial, road dust, and soil dust factors (40%). The similarity in the two approaches (modeling versus PMF) increases confidence that the dust sources are being correctly apportioned in our study. In general, emissions from the RoC for all pollutants presented a smaller contribution to local concentrations than Québec or US emissions (**Fig. 6**). Therefore, GEOS-Chem emphasized the important role of transboundary input from the US during our sampling period. $PM_{2.5}$ can be reduced through local mitigation strategies, but the effectiveness of such strategies will be limited without updating international agreements to further reduce transboundary pollution.

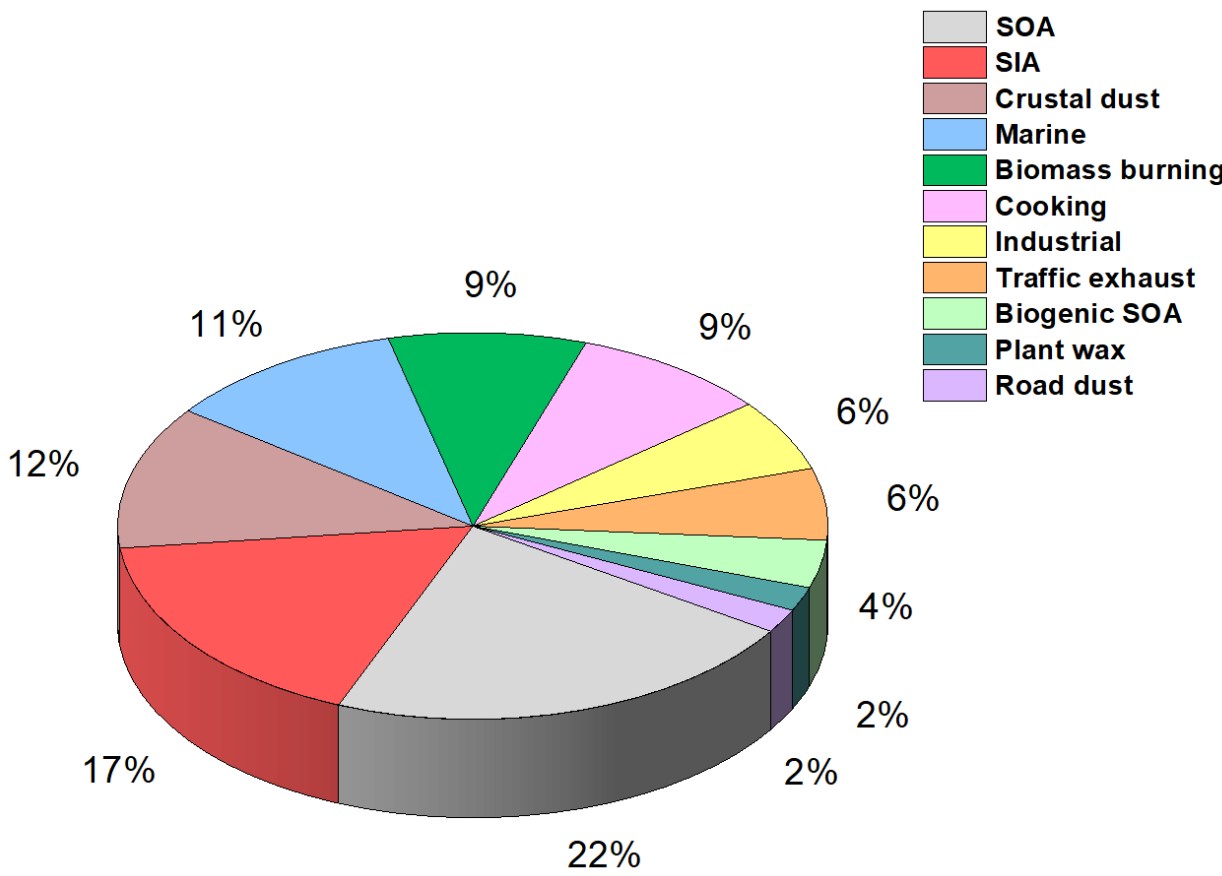

**Figure 5.** Contributions of the eleven identified sources to the total PM$_{2.5}$ mass.

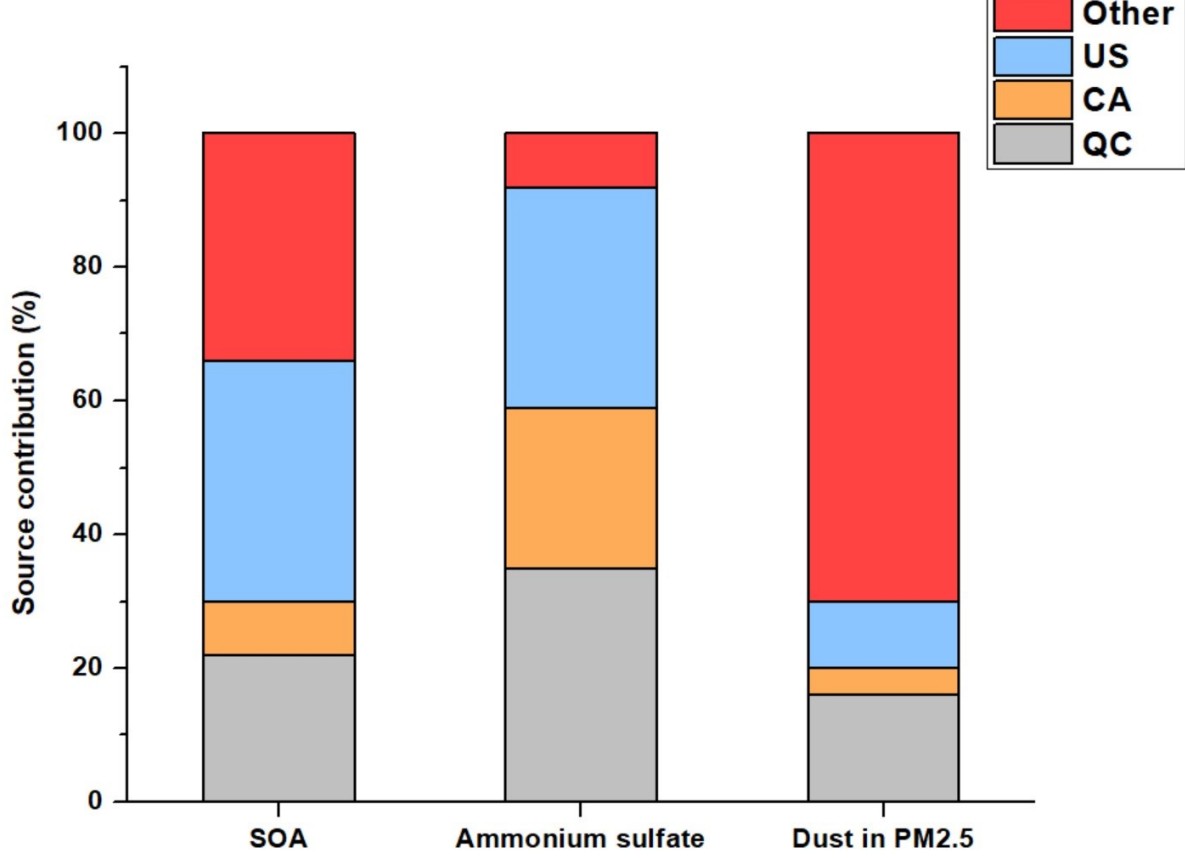

 **Figure 6.** Contribution of anthropogenic emissions from three different source regions, namely Québec (QC), United States (US) and the rest of Canada (CA) to concentrations of SOA, ammonium sulfate, and dust in $PM_{2.5}$ between August and November 2020, as predicted by GEOS-Chem.

### 3.7 Health risk assessment of $PM_{2.5}$ elements

The non-carcinogenic health risks from airborne elements through inhalation, ingestion and dermal contact were estimated. The HI exhibited the same trend for both children and adults and followed a decreasing order of Co > Mn > Pb > Cu > Cd > Ni > Cr(VI) > Sb > Zn> Al > Fe > V. Among the three exposure pathways, inhalation contributed the most to the total non-carcinogenic risk ($HI_{total}$). Overall, inhalation contributed 98% for adults and 97% for children to $HI_{total}$, ingestion contributed 1% (adults) and 2% (children) while dermal absorption was the remainder. Co was the largest contributor to the $HI_{inhalation}$, with a contribution of 61% for both adults and children. An HI value higher than one implies that adverse effects other than cancer such as cardiovascular and respiratory diseases are expected (Fadel et al., 2022a). In this study, HI and $HI_{total}$ ($HI_{total}$ =

0.24 for adults and 0.36 for children) were below the level of 1, highlighting limited non-carcinogenic health hazards from PM$_{2.5}$-metals.

650 The carcinogenic risk of each carcinogenic metal (i.e., Cr(VI), Co, Ni, V, Cd and Pb) from the three exposures pathways was also calculated. Inhalation was the exposure pathway with the highest cancer risk to which 99% and 98% of the overall carcinogenic risk (CR$_{total}$) for adults and children was ascribed, respectively. According to the USEPA, a cancer risk value between $10^{-6}$ (one additional case per one million people) and $10^{-4}$ (one in a ten thousand) indicates that the carcinogenic risk is considered as tolerable, while a value higher than $10^{-4}$ indicates that a serious risk of cancer exists (Bari and

655 Kindzierski, 2016; Dahmardeh Behrooz et al., 2021).

In this study, the carcinogenic risk of V, Ni, Cd and Pb was between $1.18 \times 10^{-8}$ and $6.11 \times 10^{-7}$, which is lower than $10^{-6}$. However, Co and Cr(VI) presented a cancer risk higher than $10^{-6}$ (**Fig. 7**), highlighting that more attention should be given to these trace metals. Based on PMF analysis, Co was associated with industrial emissions and the correlation between Co and Cr (R=0.79, *p<0.01*) (section 3.4) suggests a common source from industrial emissions for these two trace elements (Cr

660 was not added in the PMF analysis because it was not well modeled). The sum of the risk levels posed by the six metals was $1.77 \times 10^{-5}$ and $5.87 \times 10^{-6}$ for adults and children, respectively, which is between the range of $10^{-6}$ - $10^{-4}$, indicating that the carcinogenic risk is considered as tolerable.

These results show that even though industrial emissions presented a very small contribution in term of mass (0.26 µg/m$^3$), the health risks associated with this source cannot be fully determined by the PM$_{2.5}$ mass concentration alone, and the trace

665 elements emitted from industrial sources source were found to have a potential health risk. Thus, this study highlights that mitigation strategies should also prioritize reduction in metals in addition to PM.

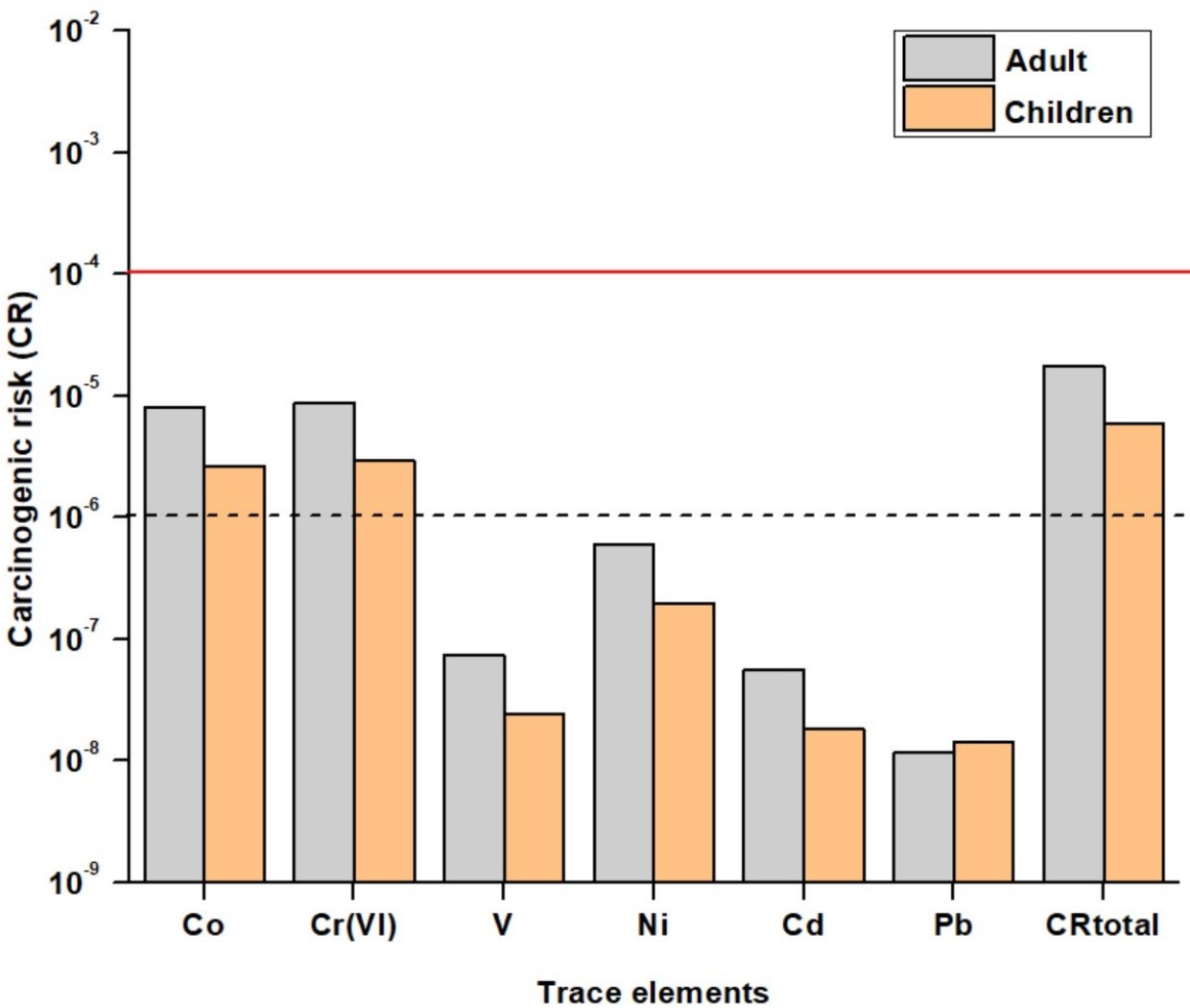

**Figure 7.** The carcinogenic risk ($CR_{total}$) from trace metals in $PM_{2.5}$. A cancer risk value between $10^{-6}$ and $10^{-4}$ indicates that the CR is considered tolerable while a value higher than $10^{-4}$ indicates that cancer risk exists seriously.

## 4 Conclusion

This work examines $PM_{2.5}$ sources in Montréal using detailed chemical speciation data collected over a 3-month period (August-November 2020). The chemical composition data included concentrations of the major components of $PM_{2.5}$ such as 675 OC, EC, water-soluble ions, and elements. These species, along with a large suite of organic tracers were used as inputs in a source apportionment model (PMF) to identify and quantify the sources of $PM_{2.5}$. In Canada, the NAPS program only provides data on organic compounds that can be measured by ion chromatography, which limits the available measurements

to a small subset of polar organic compounds. Performing PMF analysis without organic species, or only a few polar organics, not only over- and underestimates some sources but also neglects some sources (Fakhri et al., 2023). On the other hand, performing PMF analysis with only organic species is valuable for understanding organic aerosol chemistry, but it neglects important sources that contribute to the PM mass such as secondary sulfate, secondary nitrate, sea salt and crustal dust. This source apportionment study, which examined the main contributing sources to $PM_{2.5}$ using a larger suite of organic molecular markers than other Canadian source apportionment studies, is the first of its sort in Canada. Furthermore, a focus was on quantifying previously unresolved sources of $PM_{2.5}$ through the inclusion in the PMF analysis of additional organic molecular markers beyond those measured typically by the Canadian government's National Air Pollution Surveillance Program (NAPS). The organic species included in the PMF model from the GC-MS analyses were namely, 6 n-alkanes, 2 fatty acids, 1 dicarboxylic acid, 2 biogenic secondary organic aerosols (SOA) tracers and hopane. This study demonstrates that having a small set of speciated organic tracers included in PMF input matrices is beneficial for understanding the sources of $PM_{2.5}$ in Canada.

SOA and SIA were major sources and constituted 39% of the measured $PM_{2.5}$ mass. The local primary anthropogenic sources, namely traffic exhaust, road dust, industrial and cooking emissions contributed to 23% of the measured $PM_{2.5}$ mass. These sources along with crustal dust and biomass burning represented the total primary aerosol and accounted altogether for 44% of $PM_{2.5}$. According to the chemical transport model GEOS-Chem, both local (from Québec) and transboundary pollution (from the United States) contribute to the observed concentrations of SOA and SIA in Montréal, but the transboundary contribution is greater than the local contribution, indicating the need to update international agreements to further limit transboundary pollution.

One of the novel aspects of the present study was the inclusion of specific organic tracers which allowed the identification of 4 sources in addition to those that are usually identified using chemical speciation datasets from government monitoring alone. Specifically, these sources (tracers) are plant wax (high MW n-alkanes), BSOA ($\alpha$-pinene oxidation products), secondary organic aerosols (dicarboxylic acids) and cooking emissions (fatty acids). Moreover, the distinction between exhaust and non-exhaust vehicular emissions was achieved by incorporating two source-specific organic tracers in the PMF model, namely n-alkanes and hopane.

Evaluation of the health risk associated with exposure to metals revealed that Co and Cr(VI) presented a cancer risk (CR) higher than $10^{-6}$, highlighting that more attention should be given to these trace metals, which originate principally from industrial emissions according to the PMF analysis. While industrial emissions are the dominant source of Co and Cr(VI), they only contribute a small amount to the total PM2.5 (6%) indicating that the prioritization of sources and sectors for mitigation strategies will be different when considering the concentrations of individual contaminants or total $PM_{2.5}$ concentration.

*Data availability:* Data used in this study can be accessed here: https://doi.org/10.5683/SP3/96IVPX. More details on the analyses are available upon request to the contact author Nansi Fakhri (nansi.fakhri@umontreal.ca).

*Author contributions.*

**Nansi Fakhri:** field campaign and collection of the filters; chemical characterization of the collected filters; analyzing the data and writing the manuscript, **Robin Stevens:** application of GEOS-chem software; review and editing the manuscript, **Arnold Downey:** chemical characterization of the collected filters by ICP-MS, **Konstantina Oikonomou**: resources, **Jean Sciare:** resources, review and editing the manuscript, **Patrick L. Hayes**: supervision, review and editing the manuscript, project administration and resources, **Charbel Afif:** supervision, review and editing the manuscript.

*Competing interests.* The authors declare that they have no conflict of interest.

*Acknowledgement.*

PLH and NF acknowledge support from the Natural Science and Engineering Research Council of Canada (NSERC) Discovery Grant Program (RGPIN/05002-0214), Canada Foundation for Innovation Grant (CFI, Leaders Opportunity Fund Projects, grant number 32277) and the Ministère de l'Environnement et de la Lutte contre les changements climatiques (MELCC). NF also acknowledges a scholarship from the *Centre de recherche en écotoxicologie du Québec* (EcotoQ), a strategic cluster funded by the *Fonds de recherche du Québec – Nature et technologies*. This publication has been produced within the framework of the EMME-CARE project, which has received funding from the European Union Horizon 2020 Research and Innovation Program (under grant agreement no. 856612) and the Cyprus Government. The research was also enabled in part by support provided by Calcul Québec (calculquebec.ca) and the Digital Research Alliance of Canada (alliancecan.ca). This work contains modified Copernicus Atmosphere Monitoring Service Information [2020]. Neither the European Commission nor ECMWF is responsible for any use that may be made of the information it contains.

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
