# Peer review of "Source Apportionment of PM2.5 in Montréal, Canada and Health Risk Assessment for Potentially Toxic Elements"

_EGUsphere, 2023_

## Author Comment (AC1)

We would like to thank the reviewer for their comments that helped to improve the manuscript. Hereafter, we have responded to the various comments. Some of the comments were directly answered within the text of the manuscript and the modified text is reproduced below in quotation marks.

**Reviewer comments:**

**Reviewer #1:** The manuscript presents a source apportionment study in the city of Montreal, Canada and looks at associated health risks. Daily filter samples were used during a 3-month period and analysed for a comprehensive chemical composition, including a number of organic molecular markers to better identify sources. Further the study utilises a chemical transport model to identify source regions and evaluates the health risks of measured components.

In my opinion the manuscript represents a good contribution to existing literature and the topic is relevant. The scientific quality is sound, and the analysis has been performed and presented with care. The structure of the manuscript, the results and presentation are clear. Thus, I believe, the manuscript is worth publication in ACP/EGUsphere, however, I do have some comments below:

**Major comments:**

1. **The study took place during the Covid Pandemic in 2020 but there is no mention of what impact this may have had on the outcome of the study. Even though, from what I can find, Montreal was not in a lockdown during that period, activities will have altered and thus might have influenced local and transboundary pollution. I think this point needs to be addressed.**

**Answer:** We agree with the reviewer on the importance of mentioning that this study took place during the COVID-19 pandemic in 2020. During our sampling period, Montréal was in partial lockdown where public spaces (e.g., bars, gyms, cinemas, museums, libraries and casinos) were closed due to the possibility of a second wave of the COVID-19

pandemic. We have mentioned in **lines 300-302** in the manuscript that the concentrations of $PM_{2.5}$ in 2020 were not too different in comparison with the previous years (2018 and 2019) for data for the same dates of the year (13 August to 11 November) and locations. Thus, the characteristics of the sources of $PM_{2.5}$ identified in this study are likely to be similar to other years. The comment of the referee was taken into consideration and the following paragraph was added in the text:

"It is important to mention that during our sampling period, Montréal was in partial lockdown where public spaces (e.g., bars, gyms, cinemas, museums, libraries and casinos) were closed due to the possibility of a second wave of the COVID-19 pandemic. Primary and some secondary schools were opened during that period. While these considerations suggest that the results presented here are also applicable to pre-and post-pandemic conditions, further studies are needed before generalizing the results of this study other periods. "

2. **Summary needs to be clearer in what species have been used for source apportionment. The sentence starting "this source apportionment study, which examined…" (line 15) sounds like the large suit of organic markers are the chemicals used for source apportionment. This section needs reworking to be clearer.**

**Answer:** The comment of the reviewer was taken into consideration and the paragraph was updated in the text:

"This source apportionment study, which examined the main contributing sources to $PM_{2.5}$ using a larger suite of organic molecular markers than other Canadian studies, is the first of its sort in Canada. A focus of this study was on quantifying previously unresolved sources of $PM_{2.5}$ through the inclusion in the PMF analysis of additional organic molecular markers beyond those measured typically by the Canadian government's National Air Pollution Surveillance Program (NAPS). The organic species included in the PMF model were comprised of six n-alkanes, two fatty acids, one dicarboxylic acid, two biogenic secondary organic aerosols (SOA) tracers and hopane."

3. **Section 2.3 Enrichment factors and respective results: The enrichment factors were calculated with Al as reference element, however, later in the text (line 442, p19) there is mention of Aluminium production. Will this impact the Al concentrations within Montreal and thus is Al an appropriate reference element?**

**Answer:** The enrichment factor (EF) is a qualitative method that is used to differentiate between natural and anthropogenic sources of metals in the samples. EF (Eq. 1) is defined as the ratio of the considered element concentration ($C_X$) to the reference element concentration ($C_{ref}$) in PM$_{2.5}$ divided by the same ratio for crustal material retrieved from the upper crust (Mason & Moore, 1982). Typical reference elements used in the literature are Al, Ti, and Fe (Rodriguez-Espinosa et al., 2017; Amil et al., 2016). An EF value close to 1 indicates an element originates from crustal materials while an EF higher than 10 indicates a strong anthropogenic source (Esmaeilirad et al., 2020).

$$EF= \left(\frac{C_X}{C_{ref}}\right)_{air} / \left(\frac{C_X}{C_{ref}}\right)_{crust} \qquad (Eq.\ 1)$$

We have reported in the manuscript the results using Al as a reference. We have also checked the results using Ti and Fe, and all three reference elements pointed to the same results (Fig. 1 in this authors' comment). We agree with the reviewer that we have mentioned in the manuscript (**lines 442-444**) that aluminum production and industrial processes related to metallurgy contribute to air pollution in Québec, although this part of manuscript is focused on sulphur emissions. To clarify, we have added the following sentence to the manuscript:

"Although based on the weak correlation between sulfate and Al and strong correlation of Al with crustal elements, we believe that aluminium production is not an important source of particulate aluminum at our site."

It is important to mention that in the updated version of the manuscript, we have decided to remove the section concerning the EF based on the recommendation of the second reviewer.

[Figure]

**Fig. 1:** Enrichment factor of selected elements in PM2.5 using Al, Ti and Fe as a reference element. An EF higher than 10 indicates an anthropogenic source.


With respect to point (b), all species included in the PMF analysis were above DL for all samples except for some elements (between 1 and 12% were below the DL). The following table (Table 1) was added in the supplementary information of all the species included in PMF and the percentage below LOD.

**Table 1:** Species included in PMF.

| Species | % of data below the DL | Species | % of data below the DL |
|---|---|---|---|
| OC | - | Levoglucosan | - |
| EC | - | 7α[H]-21β[H]-Hopane | - |
| $Na^+$ | - | Hexadecanoic acid | - |
| $Cl^-$ | - | Octadecanoic acid | - |
| $NH_4^+$ | - | C20 | - |
| $NO_3^-$ | - | C21 | - |
| $SO_4^{2-}$ | - | C24 | - |
| Al | 3 | C25 | - |
| Fe | 1 | C27 | - |
| Ti | - | C29 | - |
| Cu | 4 | Oxalic acid | - |
| Sb | 3 | Pinic acid | - |
| Cd | 5 | Cis-pinonic acid | - |
| Co | 12 | | |

With respect to point (c), Figure 5 was updated (Fig. 2 below).

[Figure]

**Fig. 2:** Profiles of the eleven factors identified from the PMF model. The left axis corresponds to concentration (blue bars) and the right axis percentage (orange markers). Units of concentration are ng/m$^3$

With respect to points (d) and (e), in the reviewer's comment above, the objective function Q in PMF is considered as a critical parameter. PMF minimizes it when determining factor contributions and profiles. Since the number of factors in PMF is unknown, we started with the minimum number of factors (which is 2) and we started increasing this number. To select the appropriate number of factors, different mathematical diagnostic methods were investigated such as the maximum individual mean (IM) and the maximum individual standard deviation (IS) (as described in the Supplementary Information). Graphical representations of IM and IS statistics along with the Q-value showed generally a constant decrease of their values when increasing the number of factors and a stabilization starting with the 11-factor solution, which suggests that 11 is the optimal number of solutions. However, we have also used our understanding of the probable sources impacting the sampling site and the species characteristics to choose the most suitable number of factors.

To ensure robust results in PMF, several points were taken into consideration:

- The authors ensured that the uncertainty-scaled residuals of all the species are generally normally distributed with residuals varying between -3 and +3.
- The authors checked that all the species are well modeled with high determination coefficients ($R^2$) between observed and predicted observations.
- The authors examined the $Q/Q_{exp}$ values for the different species and ensured that this value was lower than 2 as recommended in the EPA PMF manual (USEPA, 2014). For each species, the $Q/Q_{exp}$ is the sum of the squares of the scaled residuals for that species divided by the overall $Q_{exp}$ divided by the number of strong species. Thus, examining $Q/Q_{exp}$ is an efficient way to understand the residuals of a PMF solution.
- The authors compared the resulting source profiles against the literature.
- The variation of $Q/Q_{exp}$ ratio from 3 to 14 factors is now provided in the supplementary information as requested by the reviewer.
- Lastly, the robustness of the PMF solution was tested by the two-error estimation method (bootstrap and displacement) as instructed in the PMF manual to ensure the solution was stable (Table S2 in the supplementary information) (USEPA, 2014).


$$SOC = OC_{total} - EC \times \left(\frac{OC}{EC}\right)_{min} \quad (Eq.\,2)$$

In the first step, the OC/EC ratio is calculated for each sample, and (OC/EC)min is the minimum ratio observed in the samples. In this study, (OC/EC)min was 2.22. In the second step, the measured OC (OCtotal) and EC for each sample are used with the minimum to calculate the SOC following the equation above. "

[Figure]

**Fig. 3:** The temporal variation of OC/EC ratio for the sampling period.


**P18section3.5 This references the mass closure results. I think the mass closure should be mentioned in the text or even the methodology.**

**Answer:** The detailed mass closure methodology was moved from the supplementary information to the main text of the manuscript. The text moved to the main text is quoted below.

[revised manuscript text omitted]

**P20L463 onwards: the traffic exhaust factor still has some Al in it and Fe, thus Might there still be some mixing with road dust/crustal dust? Especially as the road dust has less aluminium than the traffic exhaust – see also comment about PMF in general as the factor profiles in the figure would benefit from more information.**

**Answer:** We agree with the reviewer that there may be some very small mixing of the traffic exhaust, road dust and crustal dust factors, which is a limitation of this study, but the amount of mixing is very minor and should not impact the conclusions drawn from these results. In this study, PMF allocated 76% of Fe and 68% of Al to the crustal dust factor. In comparison, if we look closely at the results (Fig. 4 below), only 2% of Fe was

in the road dust factor while the amount of Al was 4%. Moreover, Fe was 2% in the traffic exhaust factor and Al was 6%. These are therefore very small amounts which may indicate a slight mixing of the factors. However, it is also possible that these metals are actually associated with the identified sources. Previous literature has found Fe and Al-containing particles in vehicle exhaust (Golokhvast et al., 2015; Wang et al., 2021). It is also logical that road dust would contain some crustal elements.

[Figure]

**Fig. 4 (Part 1):** Profiles of the factors identified from the PMF model. Loading (in percentage) is indicated on the vertical axes.

[Figure]

**Fig. 4 (Part 2):** Profiles of the factors identified from the PMF model. Loading (in percentage) is indicated on the vertical axes.

**P20L481 onwards: Similar to the previous comment, I wonder who there is some Cu, Sb, Fe in the biogenic SOA – is there still some mixing? I guess from the supplementary material it sounds like a higher solution split the factors too far, so maybe just a comment or a reference that may have experienced the same issues would be useful.**

**Answer:** We agree with the reviewer that when looking at the biogenic SOA, we notice that there is some Cu, Sb and Fe (below 10%). If we look closely at this factor, the biogenic SOA factor was identified based on high loading of pinic acid (75%) and pinonic acid (66%). On the other hand, Fe was only 5%, Sb was 6% and Cu was 5%. Fe was allocated in much higher proportions to the crustal dust factor, Sb to the road dust factor and Cu to

the industrial factor. Our PMF analysis is consistent with a study reported by Fadel et al. (2023). Fadel and coworkers have also included biogenic SOA tracers in the PMF analysis and in their biogenic SOA profile (Fig. 5) one also notices small amounts of metals/elements.

[Figure]

**Fig. 5:** PM2.5 profiles calculated via PMF in Fadel et al. (2023)

Reference:

Fadel, M., Courcot, D., Seigneur, M., Kfoury, A., Oikonomou, K., Sciare, J., Afif, C., 2023. Identification and apportionment of local and long-range sources of PM2.5 in two East-Mediterranean sites. Atmos. Pollut. Res. 14, 101622. https://doi.org/10.1016/j.apr.2022.101622

---

## Author Comment (AC2)

We would like to thank the reviewer for their comments that helped to improve the manuscript. Hereafter, we have responded to the various comments. Some of the comments were directly answered within the text of the manuscript and the modified text is reproduced below in quotation marks.

**Reviewer #2: Source Apportionment of PM2.5 in Montreal, Canada and Health Risk Assessment for Potentially Toxic Elements.**

**This work dealt with an analysis of PM2.5 collected in a sampling campaign that lasted roughly 3 months (actually 80 days) in Montreal, a populous city in Quebec, Canada. The analysis involved factor analytical source apportionment with positive matrix factorization, use of enrichment factors, a chemical transport modelling exercise with GEOS-CHEM and a health risk assessment of components of the sampled PM. The authors make some statements either implicitly or explicitly that can be considered as the main results/conclusions of the work:**

**1.   Their chemical analyses are an exhaustive characterization of PM2.5. By contrast, the analyses done by Environment Canada within the National Air Pollution Surveillance (NAPS) framework is inadequate in fully characterizing the organic species in PM2.5.**

**2.   The inclusion of their chosen tracers helps them identify and distinguish certain factors in their PMF analyses. Implying that these factor identifications would not have been possible/successful otherwise for the 11 factors found.**

**3.   Certain factors with low mass are likely more critical from their health risk analyses perspective. Thus, implying that reductions in PM2.5 mass concentrations do not necessarily translate to healthier air quality.**

**4.   Their GEOS-CHEM analyses results for SOA, ammonium sulphate and 'Dust in PM2.5' are said to show that SOA and ammonium sulphate have substantial origins in the US. 'Other' sources dominate the 'Dust in PM2.5'.**

**To start, the chemical analyses is not a complete characterization of particulate organic matter. Thus, to suggest that this study in some way improves on the NAPS method for organic PM is a stretch. There are entire compound classes of organic compounds that are missing in the proposed approach ranging the entire gamut of non-polar to polar compounds. Also, the practicality of perpetually running a chemical laboratory for exhaustive characterization of all organic compound**

**classes for air monitoring locations across an entire country is glossed over by the authors likely due to the fact that this study is an intensive 90-day sampling campaign, where it may be possible to analyze some more compounds than the standard NAPS protocol. There is always a trade-off between the frequency of analyses and how many components can be reliably analyzed. For long term monitoring, determining all organic particulate matter components is unrealistic for analytical laboratories, even if it is feasible for short-term campaigns such as this study of 80 near-consecutive days.**

**Answer:** We agree with the reviewer that it is not feasible to perform an exhaustive chemical characterization of PM within the framework of a regular monitoring program. The aim of this study is to suggest adding one more instrument on top of the analyses already done by Environment Canada within the National Air Pollution Surveillance (NAPS) program, namely gas chromatography-mass spectrometry (GC-MS) to identify a small subset of organic tracers in PM$_{2.5}$. We are only proposing a few organic tracers that are particularly useful in source apportionment studies for better PMF results. Therefore, we entirely agree with the reviewer that for long term monitoring, determining all organic PM components is unrealistic for analytical laboratories, however, we are pushing towards determining some organic tracers.

Many articles in the literature are using PMF with little or no information on the organic composition of PM$_{2.5}$ (Alwadei et al., 2022; Lee et al., 2022; Han et al., 2022; Diao et al., 2022; Camilleri et al., 2022; Guo et al., 2021; Duan et al., 2021; Manousakas et al., 2020; Zhang et al., 2020; Park et al., 2019; Soleimanian et al., 2019; Galon-Negru et al., 2019; Luo et al., 2018). A smaller number of PMF studies have focused on organic molecular markers (Gadi et al., 2019; Shivani et al., 2019; Williams et al., 2010; Gupta et al., 2018) or combined both organic and inorganic tracers (Lu et al., 2018; Wong et al., 2019; Lv et al., 2021). In Canada, the NAPS program only provides data on organic compounds that can be measured by ion chromatography, which limits the available measurements to a small subset of polar organic compounds.

While performing PMF analysis with only organic species is valuable for understanding OA chemistry, it neglects important sources that largely contribute to the PM mass such as secondary sulfate, secondary nitrate, sea salt and crustal dust due to the absence of a specific organic tracer. On the other hand, performing PMF analysis without organic

species results, or only a few polar organics, not only in over- and underestimates some sources but also neglects some sources (Wang et al., 2019; Fakhri et al., 2023). This source apportionment study, which examined the main contributing sources to PM$_{2.5}$ using a larger suite of organic molecular markers than other Canadian studies, is the first of its sort in Canada. A focus of this study was on quantifying previously unresolved sources of PM$_{2.5}$ through the inclusion in the PMF analysis of additional organic molecular markers beyond those measured typically by the Canadian government's National Air Pollution Surveillance Program (NAPS). The organic species included in the PMF model from the GC-MS analyses were namely, 6 n-alkanes, 2 fatty acids, 1 dicarboxylic acid, 2 biogenic secondary organic aerosols (SOA) tracers and hopane. In this paper, we are demonstrating that having a small set of speciated organic tracers included in PMF input matrices would be beneficial for understanding the sources of PM$_{2.5}$ in Canada. We would kindly suggest that their identification by GC-MS within the NAPS program would not be an unreasonable expansion of the program. Even if GC-MS measurements were performed for a subset of NAPS stations and a subset of days (e.g., once in 6 days), which would lessen the burden on the NAPS program, such data could be included in future PMF analyses.

References

Alwadei, M., Srivastava, D., Alam, M.S., Shi, Z., Bloss, W.J., 2022. Chemical characteristics and source apportionment of particulate matter (PM2.5) in Dammam, Saudi Arabia: Impact of dust storms. Atmospheric Environment, X 14. https://doi.org/10.1016/j.aeaoa.2022.100164

Camilleri, R., Vella, A.J., Harrison, R.M., Aquilina, N.J., 2022. Source apportionment of indoor PM2.5 at a residential urban background site in Malta. Atmospheric Environment. 278, 119093. https://doi.org/10.1016/j.atmosenv.2022.119093

Diao, Y., Liu, A., Hu, Q., Yang, M., Zhao, T., Cui, Y., Shi, S., Kong, X., 2022. Characteristics of chemical composition and source apportionment of PM2.5 during a regional haze episode in the yangtze river delta, china. Frontiers in Environmental Science. 10, 1–13. https://doi.org/10.3389/fenvs.2022.1027397

Duan, X., Yan, Y., Li, R., Deng, M., Hu, D., Peng, L., 2021. Seasonal variations, source apportionment, and health risk assessment of trace metals in PM2.5 in the typical industrial city of changzhi, China. Atmos. Pollut. Res. 12, 365–374. https://doi.org/10.1016/j.apr.2020.09.017

Fakhri, N., Fadel, M., Pikridas, M., Sciare, J., Hayes, P.L., Afif, C., 2023. Source apportionment of PM2.5 using organic / inorganic markers and emission inventory evaluation in the East Mediterranean-Middle East city of Beirut. Environ. Res. 223, 115446. https://doi.org/10.1016/j.envres.2023.115446

Gadi, R., Shivani, Sharma, S.K., Mandal, T.K., 2019. Source apportionment and health risk assessment of organic constituents in fine ambient aerosols (PM 2.5 ): A complete year study over National Capital Region of India. Chemosphere 221, 583–596. https://doi.org/10.1016/j.chemosphere.2019.01.067

Galon-Negru, A.G., Olariu, R.I., Arsene, C., 2019. Size-resolved measurements of PM2.5 water-soluble elements in Iasi, north-eastern Romania: Seasonality, source apportionment and potential implications for human health. Sci. Total Environ. 695, 133839. https://doi.org/10.1016/j.scitotenv.2019.133839

Guo, Q., Li, L., Zhao, X., Yin, B., Liu, Y., Wang, Xiaoli, Yang, W., Geng, C., Wang, Xinhua, Bai, Z., 2021. Source apportionment and health risk assessment of metal elements in PM2.5 in central liaoning's urban agglomeration. Atmosphere (Basel). 12. https://doi.org/10.3390/atmos12060667

Gupta, S., Gadi, R., Sharma, S.K., Mandal, T.K., 2018. Characterization and source apportionment of organic compounds in PM10 using PCA and PMF at a traffic hotspot of Delhi. Sustainable Cities and Society. 39, 52–67. https://doi.org/10.1016/j.scs.2018.01.051

Han, S., Joo, H., Song, H., Lee, S., Han, J., 2022. Source Apportionment of PM2.5 in Daejeon Metropolitan Region during January and May to June 2021 in Korea Using a Hybrid Receptor Model.

Lee, Y.S., Kim, Y.K., Choi, E., Jo, H., Hyun, H., Yi, S.M., Kim, J.Y., 2022. Health risk assessment and source apportionment of PM2.5-bound toxic elements in the industrial city of Siheung, Korea. Environmental Science and Pollution Research. 29, 66591–66604. https://doi.org/10.1007/s11356-022-20462-0

Lu, X., Wang, Y., Li, J., Shen, L., & Fung, J. C. H. 2018. Evidence of heterogeneous HONO formation from aerosols and the regional photochemical impact of this HONO source. Environmental Research Letters, 13(11). https://doi.org/10.1088/1748-9326/aae492

Luo, Y., Zhou, X., Zhang, J., Xiao, Y., Wang, Z., Zhou, Y., Wang, W., 2018. PM 2.5 pollution in a petrochemical industry city of northern China: Seasonal variation and source apportionment. Atmos. Res. 212, 285–295. https://doi.org/10.1016/j.atmosres.2018.05.029

Lv, L., Chen, Y., Han, Y., Cui, M., Wei, P., Zheng, M., & Hu, J. 2021. High-time-resolution PM2.5 source apportionment based on multi-model with organic tracers in Beijing during haze episodes. Science of the Total Environment, 772, 144766. https://doi.org/10.1016/j.scitotenv.2020.144766

Manousakas, M.I., Florou, K., Pandis, S.N., 2020. Source apportionment of fine organic and inorganic atmospheric aerosol in an urban background area in Greece. Atmosphere (Basel). 11. https://doi.org/10.3390/atmos11040330

Park, M. Bin, Lee, T.J., Lee, E.S., Kim, D.S., 2019. Enhancing source identification of hourly PM2.5 data in Seoul based on a dataset segmentation scheme by positive matrix factorization (PMF). Atmos. Pollut. Res. 10, 1042–1059. https://doi.org/10.1016/j.apr.2019.01.013

Shivani, Gadi, R., Sharma, S.K., Mandal, T.K., 2019. Seasonal variation, source apportionment and source attributed health risk of fine carbonaceous aerosols over National Capital Region, India. Chemosphere 237, 124500. https://doi.org/10.1016/j.chemosphere.2019.124500

Soleimanian, E., Taghvaee, S., Mousavi, A., Sowlat, M.H., Hassanvand, M.S., Yunesian, M., Naddafi, K., Sioutas, C., 2019. Sources and Temporal Variations of Coarse Particulate Matter (PM) in Central Tehran, Iran. Atmos. 10, 291. doi:10.3390/atmos10050291

Wang, Q., Huang, X.H.H., Tam, F.C.V., Zhang, X., Liu, K.M., Yeung, C., Feng, Y., Cheng, Y.Y., Wong, Y.K., Ng, W.M., Wu, C., Zhang, Q., Zhang, T., Lau, N.T., Yuan, Z., Lau, A.K.H., Yu, J.Z., 2019. Source apportionment of fine particulate matter in Macao, China with and without organic tracers: A comparative study using positive matrix factorization. Atmos. Environ. 198, 183–193. https://doi.org/10.1016/j.atmosenv.2018.10.057

Williams, B.J., Goldstein, A.H., Kreisberg, N.M., Hering, S. V., Worsnop, D.R., Ulbrich, I.M., Docherty, K.S., Jimenez, J.L., 2010. Major components of atmospheric organic aerosol in southern California as determined by hourly measurements of source marker compounds. Atmospheric Chemistry and Physics, 10, 11577–11603. https://doi.org/10.5194/acp-10-11577-2010

Wong, Y. K., Huang, X. H. H., Cheng, Y. Y., Louie, P. K. K., Yu, A. L. C., Tang, A. W. Y., Chan, D. H. L., & Yu, J. Z. 2019. Estimating contributions of vehicular emissions to PM 2.5 in a roadside environment: A multiple approach study. Science of the Total Environment, 672, 776–788. https://doi.org/10.1016/j.scitotenv.2019.03.463

Zhang, W., Liu, B., Zhang, Y., Li, Y., Sun, X., Gu, Y., Dai, C., Li, N., Song, C., Dai, Q., Han, Y., Feng, Y., 2020. A refined source apportionment study of atmospheric PM2.5 during winter heating period in Shijiazhuang, China, using a receptor model coupled with a source-oriented model. Atmos. Environ. 222. https://doi.org/10.1016/j.atmosenv.2019.117157

**Moreover, the determination of organic carbon is sufficient to account for about or more than half the mass of particulate organic matter. The prescribed remedy proposed by the authors wherein some organic compounds are individually determined is also flawed from a mass balance perspective since it leads to double-counting of organic carbon mass. Their remedy cannot be considered an exhaustive analysis of particulate organic matter but is neither insignificant enough to be harmless to an overestimation of organic mass.**

**Answer:** In the literature, mass closure is a simple model that is also applied to identify source contributions (Taiwo, 2016; Mantas et al., 2014; Genga et al., 2017; Geng et al., 2013; Chow et al., 2015; Huang et al., 2014; Cesari et al., 2018). This method allocates PM mass to sources based on types of species. For example, the contribution of crustal matter is estimated by summing the concentrations of aluminum, silicon, calcium, iron, and titanium in their oxide forms (Huang et al., 2014). Another example is the contribution of sea salt that is calculated by summing the six major ions (Sciare et al., 2005; Fakhri et al., 2023). In PMF, source identification is based on the chemical profile (e.g., high percentages of speciated tracers, ratios of one species to another such as OC/EC or V/Ni) whereas in mass closure, the species are totally attributed to a specific source. Fe is for example attributed to crustal dust in mass closure. However, PMF can allocate Fe to different factors which emit Fe such as desert dust, resuspended dust or industries (Lv et al., 2021; Acciai et al., 2017; Galon-Negru et al., 2019; Ho et al., 2018; Saraga et al., 2019). All of this is based on the chemical profile of the source which is verified through comparison with the literature.

Source profiles or chemical fingerprints refer to the average relative chemical composition of the PM deriving from a pollution source (Pernigotti et al., 2016). Several source profile databases have been created across the globe and have been compiled in the United States Environmental Protection Agency (USEPA) SPECIATE database and a European database (SPECIEUROPE). To determine the chemical source profiles, the literature has focused on specific chemical fractions or species. Although the markers are not uniquely linked with emission sources, the chemical profiles play the major role of identifying the factors.

Not all the analyzed compounds (total number of 61) could be added in the PMF model given the recommended ratio of samples to tracers of 3-to-1 (Belis et al., 2019). Thus, only selected speciated organic and inorganic species were included. Moreover, not all organic species are source markers. To select the species included in the PMF analysis, it is important to have some initial knowledge of potential sources in the studied area. Then one can use the sources' chemical profiles that can be found in the European and

American databases as well as the literature to select appropriate tracers for the PMF analysis.

In this study, the potential sources identified qualitatively (e.g., via correlations of elements) guided us in the selection of tracers to include in the PMF analysis. Many recent articles in the literature include OC, EC and organic tracers as inputs in PMF, and there is no reason this practice would lead to an overestimation of organic mass (Lv et al., 2021; Galvao et al., 2019; Gupta et al., 2018; Wong et al., 2019; Kang et al., 2018; Lu et al., 2018; Fadel et al., 2023).

The reconstructed $PM_{2.5}$ mass ($m_{chem}$) using the mass closure method is defined as the sum of organic matter (OM), EC, crustal matter, sea salt, secondary inorganic aerosol (SIA), and other elements that are not taken into account as minerals (Chow et al., 2015). Thus, the mass closure analysis does not include the organic molecular tracers since that mass is already accounted for by the OC measurements (and Conversion Factor). Thus, we respectfully conclude that our approaches do not lead to overestimation of the organic mass, contrary to what is suggested in the comment above.

References

Acciai, C., Zhang, Z., Wang, F., Zhong, Z., Lonati, G., 2017. Characteristics and source analysis of trace elements in PM2.5 in the urban atmosphere of Wuhan in spring. Aerosol Air Qual. Res. 17, 2224–2234. https://doi.org/10.4209/aaqr.2017.06.0207

Belis, C. a, Larsen, B. R., Amato, F., Haddad, I. El, Favez, O., Harrison, R. M., Hopke, P. K., Nava, S., Paatero, P., Prévôt, A., Quass, U., Vecchi, R., & Viana, M. 2019. European Guide on Air Pollution Source Apportionment with Receptor Models. JRC References Report, March, 88. https://doi.org/10.2788/9307

Cesari, D., De Benedetto, G.E., Bonasoni, P., Busetto, M., Dinoi, A., Merico, E., Chirizzi, D., Cristofanelli, P., Donateo, A., Grasso, F.M., Marinoni, A., Pennetta, A., Contini, D., 2018. Seasonal variability of PM2.5 and PM10 composition and sources in an urban background site in Southern Italy. Sci. Total Environ. 612, 202–213. https://doi.org/10.1016/j.scitotenv.2017.08.230

Chow, J.C., Lowenthal, D.H., Chen, L.W.A., Wang, X., Watson, J.G., 2015. Mass reconstruction methods for PM2.5: a review. Air Qual. Atmos. Heal. 8, 243–263. https://doi.org/10.1007/s11869-015-0338-3

Fadel, M., Courcot, D., Seigneur, M., Kfoury, A., Oikonomou, K., Sciare, J., Afif, C., 2023. Identification and apportionment of local and long-range sources of PM2.5 in two East-Mediterranean sites 14. https://doi.org/10.1016/j.apr.2022.101622

Fakhri, N., Fadel, M., Öztürk, F., Keleş, M., Iakovides, M., Pikridas, M., Abdallah, C., Karam, C., Sciare, J., Hayes, P.L., Afif, C., 2023. Comprehensive chemical characterization of PM2.5 in the large East Mediterranean-Middle East city of Beirut, Lebanon. J. Environ. Sci. 133, 118–137. https://doi.org/10.1016/j.jes.2022.07.010

Galon-Negru, A.G., Olariu, R.I., Arsene, C., 2019. Size-resolved measurements of PM2.5 water-soluble elements in Iasi, north-eastern Romania: Seasonality, source apportionment and potential implications for human health. Sci. Total Environ. 695, 133839. https://doi.org/10.1016/j.scitotenv.2019.133839

Galvao, E.S., Reis, N.C., Lima, A.T., Stuetz, R.M., D'Azeredo Orlando, M.T., Santos, J.M., 2019. Use of inorganic and organic markers associated with their directionality for the apportionment of highly correlated sources of particulate matter. Sci. Total Environ. 651, 1332–1343. https://doi.org/10.1016/j.scitotenv.2018.09.263

Geng, N., Wang, J., Xu, Y., Zhang, W., Chen, C., Zhang, R., 2013. PM2.5 in an industrial district of Zhengzhou, China: Chemical composition and source apportionment. Particuology 11, 99–109. https://doi.org/10.1016/j.partic.2012.08.004

Genga, A., Ielpo, P., Siciliano, T., Siciliano, M., 2017. Carbonaceous particles and aerosol mass closure in PM2.5 collected in a port city. Atmos. Res. 183, 245–254. https://doi.org/10.1016/j.atmosres.2016.08.022

Gupta, S., Gadi, R., Sharma, S.K., Mandal, T.K., 2018. Characterization and source apportionment of organic compounds in PM10 using PCA and PMF at a traffic hotspot of Delhi. Sustain. Cities Soc. 39, 52–67. https://doi.org/10.1016/j.scs.2018.01.051

Ho, W.Y., Tseng, K.H., Liou, M.L., Chan, C.C., Wang, C.H., 2018. Application of positive matrix factorization in the identification of the sources of PM2.5 in Taipei city. Int. J. Environ. Res. Public Health 15, 1–18. https://doi.org/10.3390/ijerph15071305

Huang, X.H.H., Bian, Q.J., Louie, P.K.K., Yu, J.Z., 2014. Contributions of vehicular carbonaceous aerosols to PM2.5 in a roadside environment in Hong Kong. Atmos. Chem. Phys. 14, 9279–9293. https://doi.org/10.5194/acp-14-9279-2014

Kang, M., Fu, P., Kawamura, K., Yang, F., Zhang, H., Zang, Z., Ren, H., Ren, L., Zhao, Y., Sun, Y., Wang, Z., 2018. Characterization of biogenic primary and secondary organic aerosols in the marine atmosphere over the East China Sea. Atmos. Chem. Phys. 18, 13947–13967. https://doi.org/10.5194/acp-18-13947-2018

Lu, Z., Liu, Q., Xiong, Y., Huang, F., Zhou, J., Schauer, J.J., 2018. A hybrid source apportionment strategy using positive matrix factorization (PMF) and molecular marker

chemical mass balance (MM-CMB) models. Environ. Pollut. 238, 39–51. https://doi.org/10.1016/j.envpol.2018.02.091

Lv, L., Chen, Y., Han, Y., Cui, M., Wei, P., Zheng, M., Hu, J., 2021. High-time-resolution PM2.5 source apportionment based on multi-model with organic tracers in Beijing during haze episodes. Sci. Total Environ. 772, 144766. https://doi.org/10.1016/j.scitotenv.2020.144766

Mantas, E., Remoundaki, E., Halari, I., Kassomenos, P., Theodosi, C., Hatzikioseyian, A., Mihalopoulos, N., 2014. Mass closure and source apportionment of PM2.5 by Positive Matrix Factorization analysis in urban Mediterranean environment. Atmos. Environ. 94, 154–163. https://doi.org/10.1016/j.atmosenv.2014.05.002

Pernigotti, D., Belis, C.A., Spanó, L., 2016. SPECIEUROPE: The European data base for PM source profiles. Atmospheric Pollution Research, 7, 307–314. https://doi.org/10.1016/j.apr.2015.10.007

Saraga, D.E., Tolis, E.I., Maggos, T., Vasilakos, C., Bartzis, J.G., 2019. PM2.5 source apportionment for the port city of Thessaloniki, Greece. Sci. Total Environ. 650, 2337–2354. https://doi.org/10.1016/j.scitotenv.2018.09.250

Sciare, J., Oikonomou, K., Cachier, H., Mihalopoulos, N., Andreae, M.O., Maenhaut, W., Sarda-Estève, R., 2005. Aerosol mass closure and reconstruction of the light scattering coefficient over the Eastern Mediterranean Sea during the MINOS campaign. Atmos. Chem. Phys. Discuss. 5, 2427–2461. https://doi.org/10.5194/acpd-5-2427-2005
Taiwo, A.M., 2016. Source apportionment of urban background particulate matter in Birmingham, United Kingdom using a mass closure model. Aerosol Air Qual. Res. 16, 1244–1252. https://doi.org/10.4209/aaqr.2015.09.0537

Wong, Y.K., Huang, X.H.H., Cheng, Y.Y., Louie, P.K.K., Yu, A.L.C., Tang, A.W.Y., Chan, D.H.L., Yu, J.Z., 2019. Estimating contributions of vehicular emissions to PM 2.5 in a roadside environment: A multiple approach study. Sci. Total Environ. 672, 776–788. https://doi.org/10.1016/j.scitotenv.2019.03.463

**Confidence in the PMF analysis itself is very low. The slope in Figure S2 shows that at any given time, their PMF analysis accounts for only 18% of observed PM2.5 mass. The authors have focused solely on the R2 metric but failed to realize that the sum of factors must account for 80 − 100 % of the measured mass (as seen from the slope) for the apportionment to be considered relevant. Based on this fact alone, this work should not be published.**

**Answer:** We would like to thank the reviewer for pointing out the error we made by adding the incorrect figure to the text. We are aware that $R^2$ and slope are essential, and we have

revised the figure in the supplementary information. In this study, the $R^2$ between the reconstructed and measured PM$_{2.5}$ mass was 0.87 and the slope was 0.90, which conforms to the requirements suggested by the reviewer.

**To discuss the extra factor identities found in this work, it is always the case that the more disparate variables added in the input matrix, the more factors will be resolved. The challenge that arises though is establishing the linkage between 'factor' and 'source'. There is nothing in this work that goes the extra step to establish the actual sources of these novel factors that the authors claimed would not have been found without their analytical method. No attempts were made at showing temporal trends or spatial apportionments. How can it be conclusively shown that some of these new factors do not represent factor splits? Plant wax, biogenic SOA may in fact be an overextraction of the same factor that has now been split into two separate factors.**

**Answer:** The authors understand the point of view of the reviewer regarding the number of factors identified by PMF. However, many recent articles that have included both inorganic and organic species in PMF have found similar results (Fakhri et al., 2023; Lv et al., 2021; Fadel et al., 2023). Generally, we think that the number of sources identified is reasonable based on three findings: (1) the number of sources identified in previous studies is consistent with the PMF analysis in the present manuscript, (2) the source profiles of the factors are similar to previously published profiles, (3) the mode results has little rotational ambiguity.

➢ *The number of sources identified in previous studies is consistent with the PMF analysis in the present manuscript.*

Many of the organic compounds we have selected for inclusion in the PMF model are well-known source tracers, and thus it is highly likely that they will improve the factor separation and therefore necessarily increase the number of identified sources. The literature usually presents source apportionment of PM$_{2.5}$ with carbonaceous matter (EC and OC) and levoglucosan as the only input data for carbonaceous and organic matter (Achilleos et al., 2016; Hassan et al., 2021; Ikemori et al., 2021; Kim et al., 2018; Theodosi et al., 2018; Yu et al., 2019). The number of identified sources in these cases varies between 6 and 9 factors. Furthermore, papers found in the literature presenting source

apportionment studies using only organic compounds led to the identification of 5 to 7 sources (Esmaeilirad et al., 2020; Gadi et al., 2019; Gupta et al., 2018). Therefore, if it is possible to identify 6 to 9 factors using almost no data on the organic composition of PM and up to 5 to 7 factors using no data on the inorganic fraction of PM, it is this reasonable to expect that the sum of these ranges, 11 to 16 factors, would be achievable when including organic and inorganic tracers. This is a very rough estimation assuming that the organic and inorganic tracers provide orthogonal information on PM sources, but nonetheless, one can see that the 11-factor solution presented in this study is consistent with the number of factors identified in previous work. In other words, some sources can only be resolved by PMF by adding organic markers for sources such as cooking emissions, plant wax emissions, biogenic secondary organic compounds, diesel combustion, gasoline combustion, etc. In our case, the organic markers helped us resolve 5 additional factors in addition to 6 factors identified based on inorganic tracers, which is entirely consistent with number of factors in the studies cited above.

➢ *The source profiles of the factors are similar to previously published profiles.*

We acknowledge that the assignment of specific sources to the obtained PMF factors is performed mainly through the factor chemical profiles and the presence of well-known tracers. In particular, factor identification was confirmed by comparison with source profiles available in the literature and in the SPECIEUROPE European database (Pernigotti et al., 2016). The cooking emissions factor for example was identified based on the contribution of hexadecanoic and octadecanoic acids. These carboxylic acids have been used in source apportionment studies to distinguish cooking activities (Gadi et al., 2019; Lv et al., 2021; Fadel et al., 2023) (**line 460-463**). Another example is the marine factor which was characterized by the ions $Na^+$, $Cl^-$ and $NO_3^-$ , as well as a $Cl^-/Na^+$ ratio below 1.8, a profile comparable to Petit et al. (2019) (**line 449-451**). The plant wax profile was identified by high loading of C27 and C20; which is similar to Fadel et al. (2023). The SOA factor was distinguished with high loading of oxalic acid, similar to Petit et al. (2019). Biomass burning was identified by high loadings of levoglucosan and an OC/EC ratio consistent with biomass burning (Fadel et al., 2023). Thus, the comment of the reviewer

was taking into consideration and these references to a similar profiles from the literature were added in the manuscript.

**➢ The model result has little rotational ambiguity**

The robustness of the PMF solution was tested by the two-error estimation method (bootstrap and displacement) as instructed in the PMF manual to ensure the solution was stable (USEPA, 2014). Of particular relevance to the reviewer's comment above, displacement (DISP) is an analysis method that helps the user understand the effects of rotational ambiguity and explores the rotational ambiguity of the solution by assessing the range of source profiles with a given increase in the Q-value (USEPA, 2014; Paatero et al., 2014). As also presented in the supplement, Table 1 below contains swap counts for the 11 factors (columns in Table 1) for several dQmax levels (rows) where dQmax is the maximum increase in Q. The second row is for dQmax = 4, the third row dQmax=8, the fourth dQmax=15 and the fifth dQmax=25. The swap counts indicate when two factors exchange identities in the PMF solution and are a key indicator of the stability of a PMF solution. Factor swaps result in the same physical model as the original solution, but the presence of factor swaps means that all intermediate solutions (i.e., mixing of two factors) must be considered as alternative solutions. Thus, swaps occurring at dQmax = 4 indicate that there is significant rotational ambiguity and that the solution is not sufficiently robust to be used (USEPA, 2014).DISP results for our results show that there are no swaps between the factors except at the highest dQmax level., indicating a solution with little rotationally ambiguity. In contrast, if factors where "split" as suggested by the reviewer one would expect a high degree of rotational ambiguity since the split factors would not be distinct and the elements in the source profiles could be exchanged with little change in Q.

**Table 1:** Displacement error estimation

[Figure]

References

Achilleos, S., Wolfson, J.M., Ferguson, S.T., Kang, C.-M., Hadjimitsis, D.G., Hadjicharalambous, M., Achilleos, C., Christodoulou, A., Nisanzti, A., Papoutsa, C., Themistocleous, K., Athanasatos, S., Perdikou, S., Koutrakis, P., Spatial variability of fine and coarse particle composition and sources in Cyprus, Atmos. Res. 169(2016), pp. 255-270.https://doi.org/10.1016/j.atmosres.2015.10.005.

Esmaeilirad, S., Lai, A., Abbaszade, G., Schnelle-Kreis, J., Zimmermann, R., Uzu, G., Daellenbach, K., Canonaco, F., Hassankhany, H., Arhami, M., Baltensperger, U., Prévôt, A.S.H., Schauer, J.J., Jaffrezo, J.L., Hosseini, V, El Haddad, I., 2020. Source apportionment of fine particulate matter in a Middle Eastern Metropolis, Tehran-Iran, using PMF with organic and inorganic markers. Sci. Total Environ. 705, 135330. https://doi.org/10.1016/j.scitotenv.2019.135330

Fakhri, N., Fadel, M., Pikridas, M., Sciare, J., Hayes, P.L., Afif, C., 2023. Source apportionment of PM 2 . 5 using organic / inorganic markers and emission inventory evaluation in the East Mediterranean-Middle East city of Beirut. Environ. Res. 223, 115446. https://doi.org/10.1016/j.envres.2023.115446

Fadel, M., Courcot, D., Seigneur, M., Kfoury, A., Oikonomou, K., Sciare, J., Afif, C., 2023. Identification and apportionment of local and long-range sources of PM2.5 in two East-Mediterranean sites 14. https://doi.org/10.1016/j.apr.2022.101622

Gadi, R., Shivani, Sharma, S.K., Mandal, T.K., Source apportionment and health risk assessment of organic constituents in fine ambient aerosols (PM2.5): A complete year study over National Capital Region of India, Chemosphere 221(2019), pp. 583-

596.https://doi.org/10.1016/j.chemosphere.2019.01.067.

Gupta, S., Gadi, R., Sharma, S.K., Mandal, T.K., Characterization and source apportionment of organic compounds in PM10 using PCA and PMF at a traffic hotspot of Delhi, Sustain. Cities. Soc. 39(2018), pp. 52-67.https://doi.org/10.1016/j.scs.2018.01.051.

Hassan, H., Latif, M.T., Juneng, L., Amil, N., Khan, M.F., Fujii, Y., Jamhari, A.A., Hamid, H.H.A., Banerjee, T., Chemical characterization and sources identification of PM2.5 in a tropical urban city during non-hazy conditions, Urban Clim. 39(2021), p. 100953.https://doi.org/10.1016/j.uclim.2021.100953.

Ikemori, F., Uranishi, K., Asakawa, D., Nakatsubo, R., Makino, M., Kido, M., Mitamura, N., Asano, K., Nonaka, S., Nishimura, R., Sugata, S., Source apportionment in PM2.5 in central Japan using positive matrix factorization focusing on small-scale local biomass burning, Atmos. Pollut, Res. 12(2021), pp. 162-172.https://doi.org/10.1016/j.apr.2021.01.006.

Kim, S., Kim, T.-Y., Yi, S.-M., Heo, J., Source apportionment of PM2.5 using positive matrix factorization (PMF) at a rural site in Korea, J. Environ. Manage. 214(2018), pp. 325-334.https://doi.org/10.1016/j.jenvman.2018.03.027.

Lv, L., Chen, Y., Han, Y., Cui, M., Wei, P., Zheng, M., Hu, J., 2021. High-time-resolution PM2.5 source apportionment based on multi-model with organic tracers in Beijing during haze episodes. Sci. Total Environ. 772, 144766. https://doi.org/10.1016/j.scitotenv.2020.144766

Paatero, P., Eberly, S., Brown, S. G., and Norris, G. A., 2014. Methods for estimating uncertainty in factor analytic solutions, Atmos. Meas. Tech., 7, 781–797, https://doi.org/10.5194/amt-7-781-2014, 2014.

Petit, J.E., Pallarès, C., Favez, O., Alleman, L.Y., Bonnaire, N., Rivière, E., 2019. Sources and geographical origins of PM10 in Metz (France) using oxalate as a marker of secondary organic aerosols by positive matrix factorization analysis. Atmosphere (Basel). 10. https://doi.org/10.3390/atmos10070370

Theodosi, C., Tsagkaraki, M., Zarmpas, P., Grivas, G., Liakakou, E., Paraskevopoulou, D., Lianou, M., Gerasopoulos, E., Mihalopoulos, N., Multi-year chemical composition of the fine aerosol fraction in Athens, Greece, with emphasis on the contribution of residential heating in wintertime, Atmos. Chem. Phys. 18(2018), pp. 14371 14391.https://doi.org/10.5194/acp-18- 4371-2018.

USEPA, 2014. EPA Positive Matrix Factorization (PMF) 5.0 Fundamentals and User Guide. U.S. Environmental Protection Agency Office of Research and Development Washington, DC 20460.

Yu, S., Liu, W., Xu, Y., Yi, K., Zhou, M., Tao, S., Liu, W., Characteristics and oxidative potential of atmospheric PM2.5 in Beijing: Source apportionment and seasonal variation, Sci. Total Environ. 650(2019), pp. 277-287. https://doi.org/10.1016/j.scitotenv.2018.09.021.

**While the GEOS-CHEM analysis is to be lauded, its value accrues when it is used in a framework that exhaustively analyzes the receptor modelling data first before a comparison can be done. PMF factor contribution results themselves are usually subjected to spatial analyses in the form of polar meteorological plots as well as air mass back trajectory analyses for both local and regional apportionments. If these were done then compared with GEOS-CHEM, more support could have been said to derive from the latter. Thus, it is hard to believe that Quebec, a province with no coal-fired power generation, is a source of more particulate secondary sulphate than the US or the rest of Canada, as seen just by relying on the GEOS-CHEM results alone. The authors are enjoined to study the use of conditional probability plots for both local (CPF) and regional (PSCF) spatial apportionments at the very minimum. For a more thorough analyses on local and regional scales, CBPF and CWT are respectively recommended.**

**Answer:** Regarding the sources of sulphate in Québec, it would be more accurate to describe the contributions from the province and the US as the same within modeling uncertainty (35% vs. 33%). Nonetheless, the reviewer's comment that this is somewhat surprising is a valid point given the lack of coal-fire power generation in Quebec. However, it is important to consider other sources of sulphate that are important regionally. Specifically, aluminum production is a major industry in Quebec that emits large amounts of $SO_2$ (NPRID, 2022). Nearly, 70% of North American aluminum is produced in Québec. In addition, other industries involving smelting and metallurgy in Québec emit $SO_2$. When also considering the recent decreased use of coal in the US (USEIA, 2022), the equal contributions of US and Quebec emissions to sulphate is reasonable.

With respect to the suggested spatial analysis of the PMF factors in the form of polar meteorological plots, we have provided below (Figure 1) pollution roses displaying the frequency of a given concentration of a factor as a function of wind direction. The wind data was taken from a nearby meteorological station at Montréal-Pierre Elliott Trudeau Airport. In general, the pollution rose plots are consistent with the identification of the

factors proposed in the manuscript. These figures will be added to the supporting information of the manuscript along with the following text.

"Pollution rose plots (Figure 1) can be used analyze the correlations between wind direction and factor concentrations by plotting in a polar graph the frequency of different concentrations of a factor as a function of wind direction. Such analyses provide information on the potential local origin of the factors.

The traffic exhaust and road dust factors show similar polar plots with the highest concentrations of these factor being observed when the wind is from the southern and western directions. The observations of high factor concentrations with winds from these directions is expected given that major highways (Autoroutes 15 and 40) are located to the west and the south of the measurement site, and winds from the south and west tend to have higher speeds facilitating transport. The road dust factor also exhibits some periods of very high concentrations when the wind is from the northeast, possibly due to the greater influence of very local emissions and surface streets.

In contrast, the biomass burning and crustal dust factors dust showed higher concentrations when winds were from the northeast. No major highways are in this direction. The biomass burning factor showed no trend with date during the campaign period. It is possible that this factor is related to certain food preparation activities such pizzerias and bagel bakeries that traditionally use wood ovens. Similarly, the crustal dust factor may be attributable to local construction activities, although further studies of the sources of these factors is needed. Interestingly, the cooking factor, unlike the biomass burning factor, shows little dependence on wind direction, which is reasonable given the measurements site is surrounded by residential neighborhoods and many restaurants.

The SIA and SOA factor both have similar dependences on wind direction with the highest concentrations tending to be observed when the wind is from the south and southeast. As already mentioned for the traffic-related factors above, winds from this direction can potentially transport aerosol and aerosol-precursors to the measurement site from major highways located to the south and southwest of the site. Alternatively, as discussed in the main text, GEOS-Chem modeling shows large transboundary contributions from the USA to these components. Thus, the wind blowing from the south may also correspond to large

scale transport from south to north that increases the transboundary contribution to the SIA and SOA factors.

Both the biogenic SOA and plant wax factors exhibit high concentrations when winds are blowing from the northwest. In this direction is a major suburb of Montréal, Town of Mont-Royal, which contains a high density of trees relative to the rest of the metropolitan area. At the same time, we note that the biogenic SOA factor reaches moderately high concentrations for almost all wind directions, suggesting the importance of regional formation, which is expected to be important for this factor.

The marine factor exhibits relatively high concentrations for multiple wind directions including from the west and southwest. Thus, the marine factor pollution rose resembles to some extent that of road dust. It is also notable that the marine factor exhibits its highest concentrations in November when minimum temperatures were below freezing, and some snowfall occurred. Thus, it is possible that is factor originates from road salt, although further work is needed to evaluate the contribution of road salt to $PM_{2.5}$ in Montréal.

Lastly, the industrial factor exhibits its highest concentration when winds are blowing from the west and north. Many major industries on the Island of Montreal are located to the northeast of the site (e.g., the Suncor Energy Refinery). Thus, the pollution rose for the industrial factor does not correspond to the location of these sources. This discrepancy may be explained by changes in wind direction upwind of the site, especially given that the distances to some of the largest potential emitters is approximately 10 km."

[Figure]

**Figure 1:** Pollution rose plots for the PMF factors showing the frequency of a given concentration as function of wind direction.

[Figure]

**Figure 1 (continued):** Pollution rose plots for the PMF factors showing the frequency of a given concentration as function of wind direction. Wind rose is shown in lower right panel.

Reference

NPRID. National Pollutant Release Inventory Dashboard, Government of Canada. Accessed August 30, 2022. https://www.canada.ca/en/environment-climate-change/services/national-pollutant-release-inventory/tools-resources-data/all-year-dashboard.html, 2022.

USEIA, 2022. U.S. energy facts explained, https://www.eia.gov/energyexplained/us-energy-facts/

**Finally, the use of enrichment factors does not belong in contemporay source apportionment studies. Enrichment factors are flawed for incontrovertible scientific reasons, e.g., see Reimann and De Caritat. Environ. Sci. Technol. 2000, 34, 5084-5091.**

**Answer:** Enrichment factor (EF) are still used in the literature before proceeding to PMF (Esmaeilirad et al., 2020; Acciai et al., 2017; Cesari et al., 2018; Li et al., 2018; Nayebare et al., 2016). These previous studies have used EF analysis along with the correlations between elements to provide qualitative details regarding the potential sources before proceeding to a more quantitative source apportionment using PMF. Nonetheless, we share the concerns of the reviewer and while our EF analysis is largely consistent with both the observed elemental correlations and our PMF analysis the section concerning the enrichment factors was removed from the manuscript.

References

Esmaeilirad, S., Lai, A., Abbaszade, G., Schnelle-Kreis, J., Zimmermann, R., Uzu, G., Daellenbach, K., Canonaco, F., Hassankhany, H., Arhami, M., Baltensperger, U., Prévôt, A.S.H., Schauer, J.J., Jaffrezo, J.L., Hosseini, V., El Haddad, I., 2020. Source apportionment of fine particulate matter in a Middle Eastern Metropolis, Tehran-Iran, using PMF with organic and inorganic markers. Sci. Total Environ. 705, 135330. https://doi.org/10.1016/j.scitotenv.2019.135330

Acciai, C., Zhang, Z., Wang, F., Zhong, Z., Lonati, G., 2017. Characteristics and source analysis of trace elements in PM2.5 in the urban atmosphere of Wuhan in spring. Aerosol Air Qual. Res. 17, 2224–2234. https://doi.org/10.4209/aaqr.2017.06.0207

Cesari, D., De Benedetto, G.E., Bonasoni, P., Busetto, M., Dinoi, A., Merico, E., Chirizzi, D., Cristofanelli, P., Donateo, A., Grasso, F.M., Marinoni, A., Pennetta, A., Contini, D.,

2018. Seasonal variability of PM2.5 and PM10 composition and sources in an urban background site in Southern Italy. Sci. Total Environ. 612, 202–213. https://doi.org/10.1016/j.scitotenv.2017.08.230

Li, P., Sato, K., Hasegawa, H., Huo, M., Minoura, H., Inomata, Y., Take, N., Yuba, A., Futami, M., Takahashi, T., Kotake, Y., 2018. Chemical Characteristics and Source Apportionment of PM 2 . 5 and Long-range Transport from Northeast Asia Continent to Niigata in Eastern Japan 938–956. https://doi.org/10.4209/aaqr.2017.05.0181

Nayebare, S.R., Aburizaiza, O.S., Khwaja, H.A., Siddique, A., Hussain, M.M., Zeb, J., Khatib, F., Carpenter, D.O., Blake, D.R., 2016. Chemical characterization and source apportionment of PM2.5 in Rabigh, Saudi Arabia. Aerosol Air Qual. Res. 16, 3114–3129. https://doi.org/10.4209/aaqr.2015.11.0658

---

## Author Response (AR2)

We would like to thank the editor for their comments that helped to improve the manuscript. Hereafter, we have responded to the various comments. The comments were directly answered within the text of the manuscript (in track changes mode and highlighted in yellow) and the modified text is reproduced below in quotation marks.

**Minor Points**

1) **It would be beneficial to add a short discussion about the advantages and disadvantages of increasing the number of tracers both from a measurement and information standpoint. Something along the lines of an abbreviated version of the discussion on page 21 and 22 of the combined response document (response to the first point of referee 2). The conclusions could be a good section for this.**

**Answer:** The comment was taken into consideration and the conclusion was updated in the manuscript (**line 687-704**):

"This work examines PM$_{2.5}$ sources in Montréal using detailed chemical speciation data collected over a 3-month period (August-November 2020). The chemical composition data included concentrations of the major components of PM$_{2.5}$ such as OC, EC, water-soluble ions, and elements. These species, along with a large suite of organic tracers were used as inputs in a source apportionment model (PMF) to identify and quantify the sources of PM$_{2.5}$. In Canada, the NAPS program only provides data on organic compounds that can be measured by ion chromatography, which limits the available measurements to a small subset of polar organic compounds. Performing PMF analysis without organic species, or only a few polar organics, not only over- and underestimates some sources but also neglects some sources (Fakhri et al., 2023). On the other hand, performing PMF analysis with only organic species is valuable for understanding organic aerosol chemistry, but it neglects important sources that contribute to the PM mass such as secondary sulfate, secondary nitrate, sea salt and crustal dust. This source apportionment study, which examined the main contributing sources to PM$_{2.5}$ using a larger suite of organic molecular markers than other Canadian source apportionment studies, is the first of its sort in Canada. Furthermore, a focus was on quantifying previously unresolved sources of PM$_{2.5}$ through the inclusion in the PMF analysis of

additional organic molecular markers beyond those measured typically by the Canadian government's National Air Pollution Surveillance Program (NAPS). The organic species included in the PMF model from the GC-MS analyses were namely, 6 n-alkanes, 2 fatty acids, 1 dicarboxylic acid, 2 biogenic secondary organic aerosols (SOA) tracers and hopane. This study demonstrates that having a small set of speciated organic tracers included in PMF input matrices is beneficial for understanding the sources of PM$_{2.5}$ in Canada."

2) **Referee 1 had several questions regarding presence of trace elements in some of the factors (pages 12, 15, 18 of the combined response document). The replies in the response document are informative and should be included in the revised manuscript (SI ok) to clarify the results for future readers.**

**Answer:** The comment was taken into consideration and the following paragraphs were added to the manuscript (**line 549-556** and **line 568-572**):

"Upon close examination of the PMF factor profiles, one notices some very small mixing of the traffic exhaust, road dust and crustal dust factors, which is a limitation of this study. However, the amount of mixing is very minor and should not impact the conclusions drawn from these results. In this study, PMF allocated 76% of Fe and 68% of Al to the crustal dust factor. In comparison, only 2% of Fe was allocated to the road dust factor while the amount of Al was 4%. Moreover, for the traffic exhaust factor, these values were 2% and 6% for Fe and Al, respectively. It is also possible that these metals are truly associated with the identified sources. Previous literature has found Fe- and Al-containing particles in vehicle exhaust (Golokhvast et al., 2015; Wang et al., 2021). It is also logical that road dust would contain some crustal elements."

"A small percentage of Cu, Sb and Fe are attributed to Biogenic SOA. Specifically, Fe, Sb, and Cu were 5%, 6%, and 5%, respectively. Fe was allocated in much higher proportion to the crustal dust factor, Sb to the road dust factor, and Cu to the industrial factor. Our PMF analysis is consistent with a study reported by Fadel et al. (2023). Fadel and coworkers also included biogenic SOA tracers in the PMF analysis and in their biogenic SOA profile one also notices small amounts of metals/elements."

**3) Sect 3.6.1: The possibility that the marine source is from road salt should be addressed in the main text not only the supplementary information. Are there any implications in road salt vs sea salt for the chemical mass closure?**

The comment was taken into consideration and the following paragraphs were added to the manuscript (**line 520-529** and **line 172-178**):

"A marine factor was characterized by the ions $Na^+$ (46%), $Cl^-$ (69%) and $NO_3^-$ (30%), contributing to 11% of the $PM_{2.5}$. The $Cl^-/Na^+$ calculated for this factor was 0.95, which is lower than the ratio of 1.80 reported for fresh sea salt and is indicative of aged sea salt (Petit et al., 2019; Seinfeld and Pandis, 2016). The presence of high nitrate loading in the profile is also consistent with the presence of aged marine salt. The observed chloride depletion is due to the reaction of nitric and sulfuric acid with NaCl particles (Seinfeld and Pandis, 2016). While the factor has been tentatively identified as "marine", there is some evidence that this factor may originate, at least partially, from road salt. The marine factor exhibits relatively high concentrations for multiple wind directions including from the west and southwest (Fig. S10), and thus, the marine factor pollution rose resembles to some extent that of road dust. It is also notable that the marine factor exhibits its highest concentrations in November when minimum temperatures were below freezing, and some snowfall occurred. Based on these findings, we suggest that further work is needed to evaluate the contribution of road salt to $PM_{2.5}$ in Montréal."

In the chemical mass closure calculation, we assume that $Na^+$ originates from sea salt, and the ratios between $Na^+$ and other ions ($SO_4^{2-}$, $Ca^{2+}$, $K^+$, and $Mg^{2+}$) in sea salt aerosol are the same as those for seawater. However, it is possible that some $Na^+$ originates from road salt, which is principally composed of NaCl, with a small amount of $CaCl_2$ (Charbonneau, 2006). In this case, the contributions of $SO_4^{2-}$, $K^+$, and $Mg^{2+}$ would be overestimated, and the true concentration of the "road/sea salt" component would be less than that calculated. In the extreme case of the component being derived entirely from road salt, the overestimation in the concentration of this component would be approximately 20%, given the preceding equations. This error is relatively small because $SO_4^{2-}$, $K^+$, and $Mg^{2+}$ have relatively small concentrations in sea salt..

**4) Sect 3.6.2: Please add to the main text a version of the text provided in the referee response document (pg 34) regarding the potential sources of sulfate within Québec and how this could explain the similarities in contribution from the US and Québec.**

The comment was taken into consideration and the following paragraph was added in the manuscript (**line 627-633**):

"Regarding the sources of sulphate in Québec, it is somewhat surprising that the contributions from the province and the US are essentially the same (35% vs. 33%) given that there are no coal-fired powerplants in Québec while coal is still used at some powerplants in the US. This finding indicates that it is important to consider other sources that contribute to sulphate regionally. Specifically, aluminum production is a major industry in Quebec that emits large amounts of $SO_2$ (NPRID, 2022). Nearly, 70% of North American aluminum is produced in Québec. In addition, other industries involving smelting and metallurgy in Québec emit $SO_2$. When also considering the recent decreased use of coal in the US (USEIA, 2022), these alternate sources of sulphate appear to be relatively important in Québec."

**Technical**

**1) Line 77: Please include the information on the specific species that were included in PMF like is already present in the abstract.**

**Answer:** The comment was taken into consideration and the sentence was updated in the text (**line 80-83**):

"One objective of this work is to investigate previously unresolved PM sources in Montréal, by using some selected organic markers, namely six n-alkanes, hopane, two fatty acids, one dicarboxylic acid, and two biogenic secondary organic aerosols tracers and hopane in the PMF model."

**2) Line 91: Please list how long samples were typically stored before analysis.**

**Answer:** The comment was taken into consideration and the following sentence was added to the manuscript (**line 98-99**):

"Collected filters were also stored at -20 °C until analysis. Organic species and elements were immediately quantified following the field campaign (i.e., within 3 months). Analyses of the water-soluble ions, sugars, OC, and EC were performed a year after the field campaign."

**3) Line 175: CF has not yet been defined. I also encourage you to consider including an equation that shows the calculation explicitly so that it is clear how CF is being used.**

**Answer:** The comment was taken into consideration and the paragraph was updated to (**line 190-196**):

"To account for unmeasured O, N, S, and H atoms in OM, the conversion factor (CF) from OC to OM was derived using the equation OM = CF×OC. The method used to calculate the CF sums all the PM components while systematically varying the OM/OC conversion (Genga et al., 2017). To find the optimal CF to calculate OM from OC, the factor was varied from 1.2 to 2.1. The Pearson correlation (R) calculated between the reconstructed $PM_{2.5}$ and the measured mass did not change significantly (0.978-0.979), but the highest correlation and the slope closest to 1 was obtained with CF=1.6. The results of chemical mass closure study are shown in Fig. S5."

**4) Figure 4: Please clarify in the caption what the percent is referring to (i.e., percent of what?).**

**Answer:** The comment was taken into consideration and the caption in the manuscript was updated to:

"Figure 4. Profiles of the eleven factors identified from the PMF model. The left axis corresponds to the concentration of each species (blue bars) and the right axis corresponds to the percentage of each species (orange markers). Units of concentration are ng/m$^3$."

**5) Line 606: Since "other" is the largest contribution, please revise to clarify that Québec represents the highest of the apportioned contribution.**

**Answer:** The comment was taken into consideration and the conclusion was updated in the manuscript (**line 634-636**):

"On the other hand, anthropogenic dust emissions from Québec presented the highest apportioned contribution to total dust concentrations among the three regions studied, and the concentrations dropped by 16% when emissions from Québec were excluded and by 10% when US emissions were excluded."

**6) Figure S6: Please include the meaning of the red box in the caption.**

**Answer:** The comment was taken into consideration and the caption in the SI was updated to:

"Fig. S6: The temporal variation of nitrate concentrations for the sampling period at the MTL site. The red box indicates the period (end of October and November) where the nitrate concentrations were higher in comparaison with the warmer months."

**7) Figure S10: Thank you for adding this figure. I think it is helpful and adds to the manuscript. However, the figure is currently very challenging to interpret. I recommend selecting different colors (please pay attention to color blind accessibility). It would be helpful to consider using a scale that can more intuitively be interpreted in terms of increasing mass.**

**Answer:** The comment was taken into consideration and Figure S10 was updated in the SI.